# Stable population structure in Europe since the Iron Age, despite high mobility

Margaret L Antonio[1†], Clemens L Weiß[2†], Ziyue Gao[3†], Susanna Sawyer[4,5†], Victoria Oberreiter[4,5], Hannah M Moots[6,7], Jeffrey P Spence[2], Olivia Cheronet[4,5], Brina Zagorc[4,5], Elisa Praxmarer[4], Kadir Toykan Özdoğan[8], Lea Demetz[4], Pere Gelabert[4], Daniel Fernandes[4,5,9], Michaela Lucci[10], Timka Alihodžić[11], Selma Amrani[12], Pavel Avetisyan[13], Christèle Baillif-Ducros[14], Željka Bedić[15], Audrey Bertrand[16], Maja Bilić[17], Luca Bondioli[18], Paulina Borówka[19], Emmanuel Botte[20], Josip Burmaz[21], Domagoj Bužanić[22], Francesca Candilio[23], Mirna Cvetko[22], Daniela De Angelis[24], Ivan Drnić[25], Kristián Elschek[26], Mounir Fantar[27], Andrej Gaspari[28], Gabriella Gasperetti[29], Francesco Genchi[30], Snežana Golubović[31], Zuzana Hukeľová[26], Rimantas Jankauskas[32], Kristina Jelinčić Vučković[33], Gordana Jeremić[31], Iva Kaić[22], Kevin Kazek[34], Hamazasp Khachatryan[35], Anahit Khudaverdyan[36], Sylvia Kirchengast[4], Miomir Korać[31], Valérie Kozlowski[37], Mária Krošláková[26], Dora Kušan Špalj[25], Francesco La Pastina[38], Marie Laguardia[39], Sandra Legrand[37], Tino Leleković[40], Tamara Leskovar[28], Wiesław Lorkiewicz[19], Dženi Los[21], Ana Maria Silva[9,41,42], Rene Masaryk[43], Vinka Matijević[22], Yahia Mehdi Seddik Cherifi[4,44,45], Nicolas Meyer[46], Ilija Mikić[31], Nataša Miladinović-Radmilović[31], Branka Milošević Zakić[47], Lina Nacouzi[48], Magdalena Natuniewicz-Sekuła[49], Alessia Nava[50], Christine Neugebauer-Maresch[51,52], Jan Nováček[53,54], Anna Osterholtz[55], Julianne Paige[56], Lujana Paraman[57], Dominique Pieri[58], Karol Pieta[26], Stefan Pop-Lazić[31], Matej Ruttkay[26], Mirjana Sanader[22], Arkadiusz Sołtysiak[59], Alessandra Sperduti[23,60], Tijana Stankovic Pesterac[61], Maria Teschler-Nicola[4,62], Iwona Teul[63], Domagoj Tončinić[22], Julien Trapp[64], Dragana Vulović[31], Tomasz Waliszewski[59], Diethard Walter[53], Miloš Živanović[65], Mohamed el Mostefa Filah[66], Morana Čaušević-Bully[67], Mario Šlaus[68], Dušan Borić[38,69], Mario Novak[15], Alfredo Coppa[4,38,70], Ron Pinhasi[4,5*], Jonathan K Pritchard[2,71*]

*For correspondence: ropinhasi@gmail.com (RP); pritch@stanford.edu (JKP)

†These authors contributed equally to this work

[1]Biomedical Informatics Program, Stanford University, Stanford, United States; [2]Department of Genetics, Stanford University, Stanford, United States; [3]Department of Genetics, University of Pennsylvania, Perelman School of Medicine, Philadelphia, United States; [4]Department of Evolutionary Anthropology, University of Vienna, Vienna, Austria; [5]Human Evolution and Archaeological Sciences, University of Vienna, Vienna, Austria; [6]Stanford Archaeology Center, Stanford University, Stanford, United States; [7]University of Chicago, Department of Human Genetics, Chicago, United States; [8]Department of History and Art History, Utrecht University, Utrecht, Netherlands; [9]CIAS, Department of Life Sciences, University of Coimbra, Coimbra, Portugal; [10]Dipartimento di Storia Antropologia Religioni Arte Spettacolo, Sapienza University, Rome, Italy; [11]Archaeological Museum Zadar, Zadar, Croatia; [12]LBEIG, Population Genetics & Conservation Unit, Department of Cellular and Molecular Biology – Faculty of Biological Sciences, University of Sciences and Technology Houari Boumediene, Algiers, Algeria; [13]National Academy of Sciences of Armenia, Institute of Archaeology and Ethnography, Yerevan, Armenia; [14]French National Institute for Preventive Archaeological Research (INRAP)/CAGT UMR 5288, Toulouse,

France; [15]Centre for Applied Bioanthropology, Institute for Anthropological Research, Zagreb, Croatia; [16]Université Gustave Eiffel – Laboratoire ACP, Paris, France; [17]Palisada Ltd, Split, Croatia; [18]Dipartimento dei Beni Culturali, Archeologia, Storia dell'arte, del Cinema e della Musica, Università di Padova, Padova, Italy; [19]Department of Anthropology, Faculty of Biology and Environmental Protection, University of Lodz, Łódź, Poland; [20]Aix Marseille Université, CNRS, Centre Camille Jullian, Aix-en-Provence, France; [21]Kaducej Ltd, Split, Croatia; [22]Faculty of Humanities and Social Sciences, University of Zagreb, Zagreb, Croatia; [23]Bioarchaeology Service, Museum of Civilizations, Rome, Italy; [24]Museo Archeologico Nazionale di Tarquinia, Direzione Regionale Musei Lazio, Rome, Italy; [25]Archaeological Museum in Zagreb, Zagreb, Croatia; [26]Institute of Archaeology, Slovak Academy of Sciences, Nitra, Slovakia; [27]Département des Monuments et des Sites Antiques - Institut National du Patrimoine INP, Tunis, Tunisia; [28]University of Ljubljana, Faculty of Arts, Department for Archaeology, Ljubljana, Slovenia; [29]Soprintendenza Archeologia, belle arti e paesaggio per le province di Sassari e Nuoro, Sassari, Italy; [30]Department of Oriental Studies, Sapienza University of Rome, Rome, Italy; [31]Institute of Archaeology Belgrade, Belgrade, Serbia; [32]Institute of Biomedical Sciences, Vilnius University, Vilnius, Lithuania; [33]Institute of Archaeology, Zagreb, Croatia; [34]Université de Lorraine, Centre de Recherche Universitaire Lorrain d' Histoire (CRULH), Nancy, France; [35]Department of Archaeologi, Shirak Centere of Armenological Studies, National Academy of Sciences Republic of Armenia, Gyumri, Armenia; [36]Institute of Archaeology and Ethnography of the National Academy of Sciences of the Republic of Armenia, Yerevan, Armenia; [37]Musée Archéologique de l'Oise, Vendeuil-Caply, France; [38]Department of Environmental Biology, Sapienza University of Rome, Rome, Italy; [39]UMR 7041 ArScAn / French Institute of the Near East, Beirut, Lebanon; [40]Archaeology Division, Croatian Academy of Sciences and Arts, Zagreb, Croatia; [41]CEF - University of Coimbra, Coimbra, Portugal; [42]UNIARQ - University of Lisbon, Lisbon, Portugal; [43]Skupina STIK Zavod za preučevanje povezovalnih področij preteklosti in sedanjosti, Ljubljana, Slovenia; [44]Cardiolo-Oncology Research Collaborative Group (CORCG), Faculty of Medicine, Benyoucef Benkhedda University, Algiers, Algeria; [45]Molecular Pathology, University Paul Sabatier Toulouse III, Toulouse, France; [46]French National Institute for Preventive Archaeological Research (INRAP), Metz, France; [47]Museum of Croatian Archaeological Monuments, Split, Croatia; [48]L'Institut français du Proche-Orient, Beirut, Lebanon; [49]Institute of Archaeology and Ethnology Polish Academy of Sciences, Centre of Interdisciplinary Archaeological Research, Warsaw, Poland; [50]Department of Odontostomatological and Maxillofacial Sciences, Sapienza University of Rome, Rome, Italy; [51]Austrian Archaeological Institute, Austrian Academy of Sciences, Vienna, Austria; [52]Institute of Prehistory and Early History, University of Vienna, Vienna, Austria; [53]Thuringia State Service for Cultural Heritage and Archaeology Weimar, Thuringia, Germany; [54]Institute of Anatomy and Cell Biology, University Medical Centre, Georg-August University of Göttingen, Göttingen, Germany; [55]Mississippi State University, Starkville, United States; [56]University of Nevada, Las Vegas, United States; [57]Trogir Town Museum, Trogir, Croatia; [58]Université Paris 1 Panthéon-Sorbonne, Paris, France; [59]Faculty of Archaeology, University of Warsaw, Warsaw, Poland; [60]Dipartimento Asia, Africa e Mediterraneo, Università degli Studi di Napoli "L'Orientale", Naples, Italy; [61]Museum of Vojvodina, Novi Sad, Serbia; [62]Department of Anthropology, Natural History Museum Vienna, Vienna, Austria; [63]Chair and Department of Normal Anatomy, Faculty of Medicine and Dentistry, Pomeranian Medical University, Szczecin, Poland; [64]Musée de La Cour d'Or, Eurométropole de Metz, Metz, France; [65]Department of Archeology, Center for Conservation and Archeology of

Montenegro, Cetinje, Montenegro; [66]Insitut d'Archeologie, University Algiers 2, Algiers, Algeria; [67]Université de Franche Comté / UMR Chrono-Environnement, Besançon, France; [68]Anthropological Centre, Croatian Academy of Sciences and Arts, Zagreb, Croatia; [69]Department of Anthropology, New York University, New York, United States; [70]Department of Genetics, Harvard Medical School, Boston, United States; [71]Department of Biology, Stanford University, Stanford, United States

**Abstract** Ancient DNA research in the past decade has revealed that European population structure changed dramatically in the prehistoric period (14,000–3000 years before present, YBP), reflecting the widespread introduction of Neolithic farmer and Bronze Age Steppe ancestries. However, little is known about how population structure changed from the historical period onward (3000 YBP - present). To address this, we collected whole genomes from 204 individuals from Europe and the Mediterranean, many of which are the first historical period genomes from their region (e.g. Armenia and France). We found that most regions show remarkable inter-individual heterogeneity. At least 7% of historical individuals carry ancestry uncommon in the region where they were sampled, some indicating cross-Mediterranean contacts. Despite this high level of mobility, overall population structure across western Eurasia is relatively stable through the historical period up to the present, mirroring geography. We show that, under standard population genetics models with local panmixia, the observed level of dispersal would lead to a collapse of population structure. Persistent population structure thus suggests a lower effective migration rate than indicated by the observed dispersal. We hypothesize that this phenomenon can be explained by extensive transient dispersal arising from drastically improved transportation networks and the Roman Empire's mobilization of people for trade, labor, and military. This work highlights the utility of ancient DNA in elucidating finer scale human population dynamics in recent history.

## Editor's evaluation

This important study provides an impressive dataset containing more than 200 novel ancient human genome sequences and a creative, robust, novel approach for studying human migration across time. The authors' conclusions are well supported by the data, and the methods used are convincing and solid. This paper will be of great interest to population geneticists and other scholars in the field of paleogenomics.

## Introduction

Ancient DNA (aDNA) sequencing has provided immense insight into previously unanswered questions about human population history. Initially, sequencing efforts were focused on identifying the main ancestry groups and transitions during prehistoric times, for which there is no written record. Recently, aDNA sampling has expanded to more recent times, allowing the study of movements of people using genetic data alongside the well-studied historical record. However, we lack a comprehensive assessment of historical genetic structure, including characterizing genetic heterogeneity and interactions across regions. Integrating historical period genetics will be instrumental to better understanding the development of European and Mediterranean population structure from prehistoric to present-day.

Prehistoric ancient genomes have allowed disentangling the movements of people and technologies across two major demographic transitions in prehistoric western Eurasia: first the farming transition ~7500 BCE (*Lazaridis et al., 2014*; *Skoglund et al., 2012*), and later the Bronze Age Steppe migrations ~3500 BCE (*Haak et al., 2015*). Over the course of generations, genetically differentiated peoples across western Eurasia came together and admixed. As a result, most present-day European genomes can be modeled as a three-way mixture of these prehistoric groups: Western Hunter-Gatherers, Neolithic farmers, and Bronze Age Herders from the Steppe (*Haak et al., 2015*; *Lazaridis et al., 2014*) with minor contributions from other groups (*Antonio et al., 2019*; *Fernandes et al., 2020*; *Lazaridis et al., 2016*; *Morrison et al., 2020*; *Mathieson et al., 2018*). These ancestry

components are present at different proportions across western Eurasia, leading to a pattern where the genetic structure of Europe mirrors its geography (*Novembre et al., 2008*).

Given that the major ancestry components of present-day west Eurasians were largely established by the end of the Bronze Age, it is unclear how and what types of demographic processes impacted the genetic make-up of western Eurasia over the last ~3000 years, from the end of the Bronze Age to present-day. Recent studies of historical period genomes from individual regions shed light on this question; they paint a picture of heterogeneity and mobility, rather than of stable population structure. In the city of Rome alone, the population was dynamic and harbored a large diversity of ancestries from across Europe and the Mediterranean from the Iron Age (~1000 BCE) through the Imperial Roman period (27 BCE-300 CE; *Antonio et al., 2019*). Historical genomes from the Iberian Peninsula also highlight gene flow from across the Mediterranean (*Olalde et al., 2019*).

These regional reports fit well with archaeological and historical records. By the Iron Age, sea travel was already common, enabling peoples from across the Mediterranean to come into contact for trade (*Abulafia, 2011*; *Broodbank, 2013*). Subsequently, the Roman Empire leveraged its organization, labor force, and military prowess to build upon existing waterways and roads throughout Europe and create a united Mediterranean for the only time in history (*Beard, 2015*; *Harper, 2017*; *Symonds, 2017*). Not only did the Empire provide a means for movement, it also provided a reason for individuals to move. Empire building activities, broadly categorized into military, labor, and trade, pulled in people and resources from inside and outside the Empire (*Scheidel, 2019*).

We sequenced 204 new historical period genomes from across Europe and the Mediterranean to more comprehensively investigate the Roman Empire's impact on the genetic landscape suggested by these regional reports of heterogeneous, mobile populations. By analyzing genetic similarities between individuals across historical Eurasia, we were able to quantify individual movements during this time. Based on population genetic simulations, we explore potential explanations of how population structure may be maintained in the face of frequent individual dispersal.

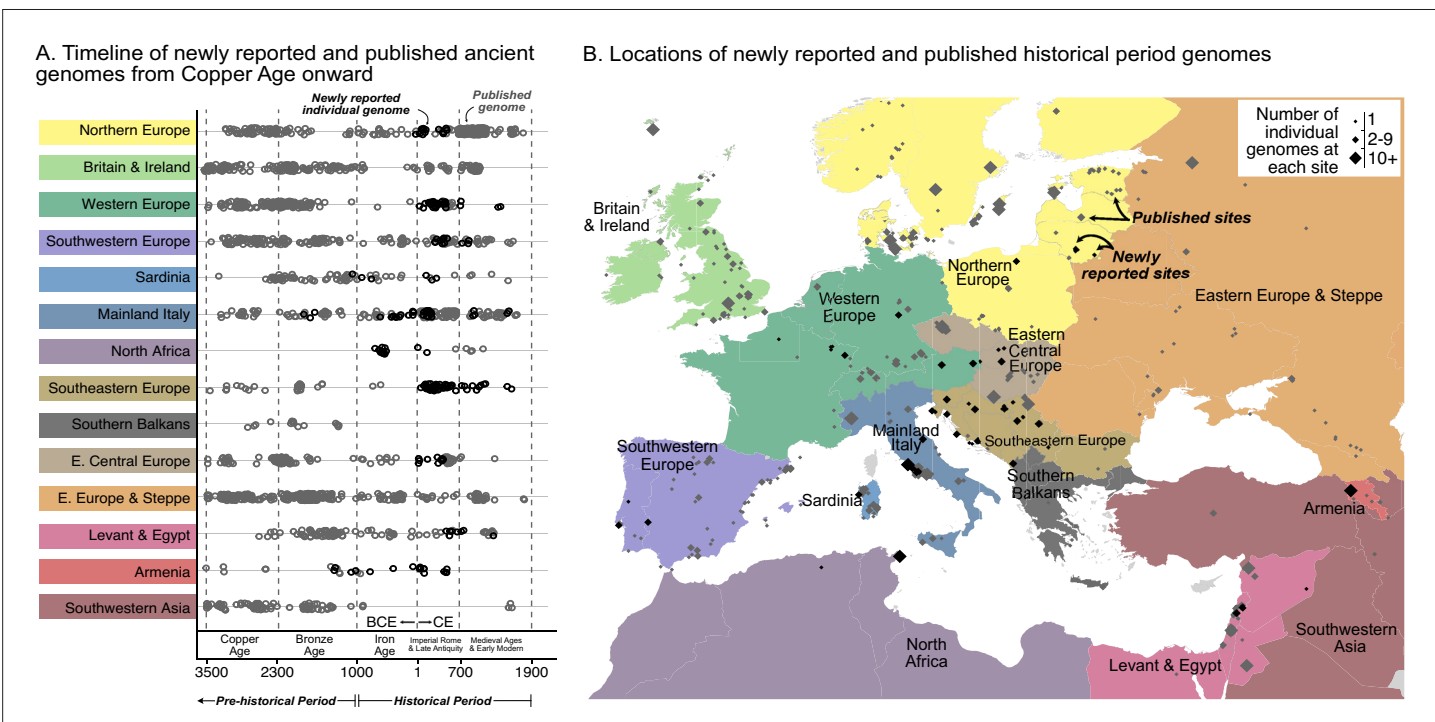

**Figure 1.** Timeline of new and published genomes. (**A**) 204 newly reported genomes (black circles) are shown alongside published genomes (gray circles), ordered by time and region (colored the same way as in B). (**B**) Sampling locations of newly reported (black) and published (gray) genomes are indicated by diamonds, sized according to the number of genomes at each location.

The online version of this article includes the following figure supplement(s) for figure 1:

**Figure supplement 1.** Detailed map of locations for newly reported samples.

## Results

### 204 new historical genomes from Europe and the Mediterranean

We collected whole genomes from 204 individuals across 53 archaeological sites in 18 countries spanning Europe and the Mediterranean (*Figure 1—figure supplement 1*), 26 of these individuals were recently reported (*Moots et al., 2022*). This collection includes the first historical genomes (Iron Age and later, i.e. after 1000 BCE) from present-day Armenia, Algeria, Austria, and France. Dates for 126 samples were directly determined through radiocarbon dating, and were used alongside archaeological contexts to infer dates for the remaining samples.

DNA was extracted from either the powdered cochlear portion of the petrous bone (n=203) or from teeth (n=1). Libraries were partially treated with uracil-DNA glycosylase (UDG) and screened for ancient DNA damage patterns, high endogenous DNA content, and low contamination. We performed whole genome sequencing to a median depth of 0.92 x (0.16x to 2.38x).

For downstream integration with published data, pseudohaploid genotypes were called for the 1240 k SNP panel (*Mathieson et al., 2015*), resulting in a median of 685,058 SNPs (167,000–1,029,345) per sample. We analyzed newly reported genomes in conjunction with 2033 present-day genomes, 1998 prehistoric genomes, and 764 published historical period genomes (*Clemente et al., 2021*; *Kovacevic et al., 2014*, *Mallick et al., 2023*; *Pagani et al., 2016*; *Saupe et al., 2021*; *Žegarac et al., 2021*, primary AADR sources cited in Materials and methods). Genomes were grouped by regions and time periods (*Figure 1*) and analyzed using principal component analysis (PCA) and *qpAdm* modeling (*Haak et al., 2015*).

### Local historical population structure varies across regions

To investigate historical population structure, we categorized the data into 14 geographical regions, split into three sub-periods of the historical period: Iron Age (1000–1 BCE), Imperial Rome & Late

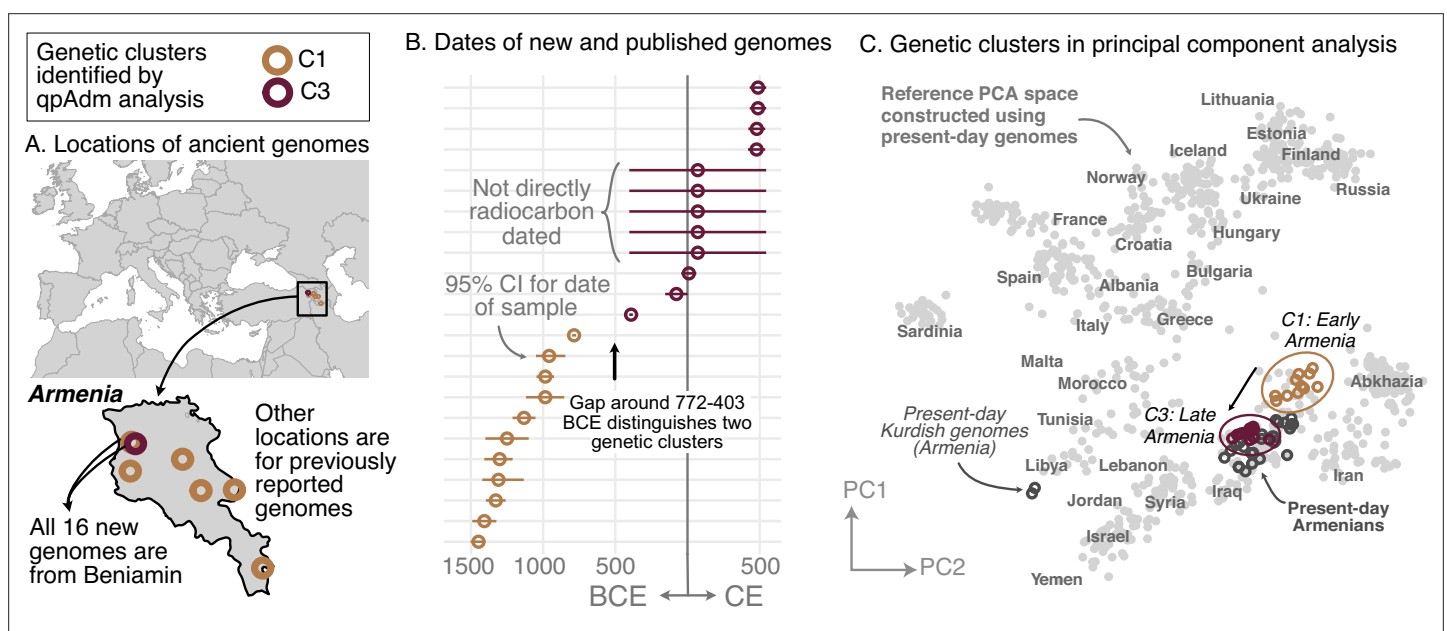

**Figure 2.** Armenia: two homogeneous genetic clusters distinguished by a temporal shift. (**A**) Sampling locations of ancient genomes (open circles) colored by their genetic cluster identified using qpAdm modeling. (**B**) Date ranges for the genomes: each line represents the 95% confidence interval for the radiocarbon date or the upper and lower limit of the inferred date, and the point represents the midpoint of that range. (**C**) Projections of the genomes onto a PCA of present-day genomes (gray points labeled by their population). Present-day genomes from Armenia are shown with dark gray open circles.

The online version of this article includes the following figure supplement(s) for figure 2:

**Figure supplement 1.** Principal component analysis of present-day genomes from Europe and the Mediterranean.

**Figure supplement 2.** Ancestry clusters identified within regions.

**Figure supplement 3.** SNP coverage comparison across cluster sizes and downstream outlier status.

Antiquity (1–700 CE), and Medieval Ages & Early Modern (700–1950 CE). We then characterized inter-individual heterogeneity within these spatio-temporal groups by examining (1) variation of projections onto a PCA space of present-day genomes (*Figure 2—figure supplement 1*), (2) genetic groups identified by *qpAdm* and clustering across time within a region, and (3) admixture modeling of genetic groups.

A majority of regions have highly heterogeneous populations in at least one historical time period (*Figure 2—figure supplement 2*). This is illustrated by both the visual spread in PCA and the genetically distinct clusters of individuals based on pairwise modeling with *qpAdm* (*Haak et al., 2015*; *Harney et al., 2021*). On average, we identified 10 genetic clusters within each region present during the historical period, with a minimum of two and a maximum of 23. With genetically similar samples grouped together, we have more power relative to individual-level analyses when performing admixture modeling on clusters of interest using *qpAdm*.

Regional vignettes reveal various patterns of historical population structure. In Armenia, for example, the population is highly homogeneous at any given time (*Figure 2*). After the Copper Age, there are two distinct genetic clusters, separated by a temporal split around 772–403 BCE (*Figure 2BC*). The earlier cluster (C1) includes newly reported samples (n=5) from Beniamin and published ones (n=6) from five other sites. This cluster cannot be modeled by any single source of ancestry using existing data. The later cluster (C3), which contains newly reported samples (n=12) from Beniamin dating between 403 BCE-500 CE, is genetically similar to present-day Armenians (excluding two Kurdish individuals; *Figure 2C*). Despite the split, there is evidence of partial continuity between the earlier and later clusters: the later (C3) can be modeled using around 50% of the earlier cluster (C1) and an additional source of Steppe ancestry. Historical genomes from Northern Europe, particularly newly reported genomes from Lithuania and Poland, exhibit a similar level of homogeneity (*Figure 2— figure supplement 2*).

In contrast to the homogeneity of the Armenian population, most of the regions, including Italy, Southeastern Europe, and Western Europe, had strikingly heterogeneous populations. Newly collected samples reinforce previous findings of high heterogeneity in Rome, including a large portion of the population having affinities for present-day Near Eastern populations (*Antonio et al., 2019*; *Posth et al., 2021*; *Figure 3—figure supplement 1*). Interestingly, Southeastern European and Western

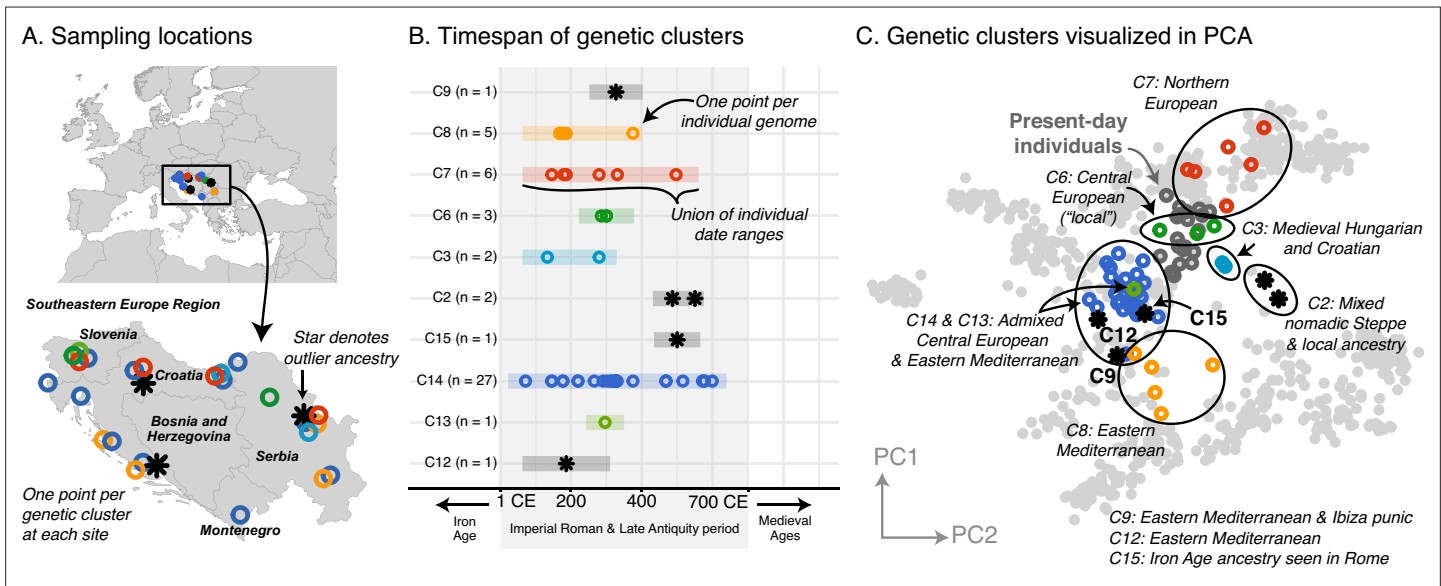

**Figure 3.** Southeastern Europe: highly heterogeneous Imperial Roman and Late Antiquity period population. (**A**) Sampling locations of genetic clusters are represented by a single point per location. Outlier ancestries are black stars, all others are open circles colored by genetic cluster. (**B**) Colored bars span the minimum and maximum of the date ranges of samples (95% confidence interval from radiocarbon dating or archaeological range). Points are the mean of an individual's date range. (**C**) Projections of the ancient genomes onto a PCA of present-day genomes (gray points). Population labels for the PCA reference space are shown in *Figure 2C*. Present-day genomes from Southeastern Europe are shown with dark gray open circles.

The online version of this article includes the following figure supplement(s) for figure 3:

**Figure supplement 1.** Population structure of Italy during the Imperial Roman and Late Antiquity period.

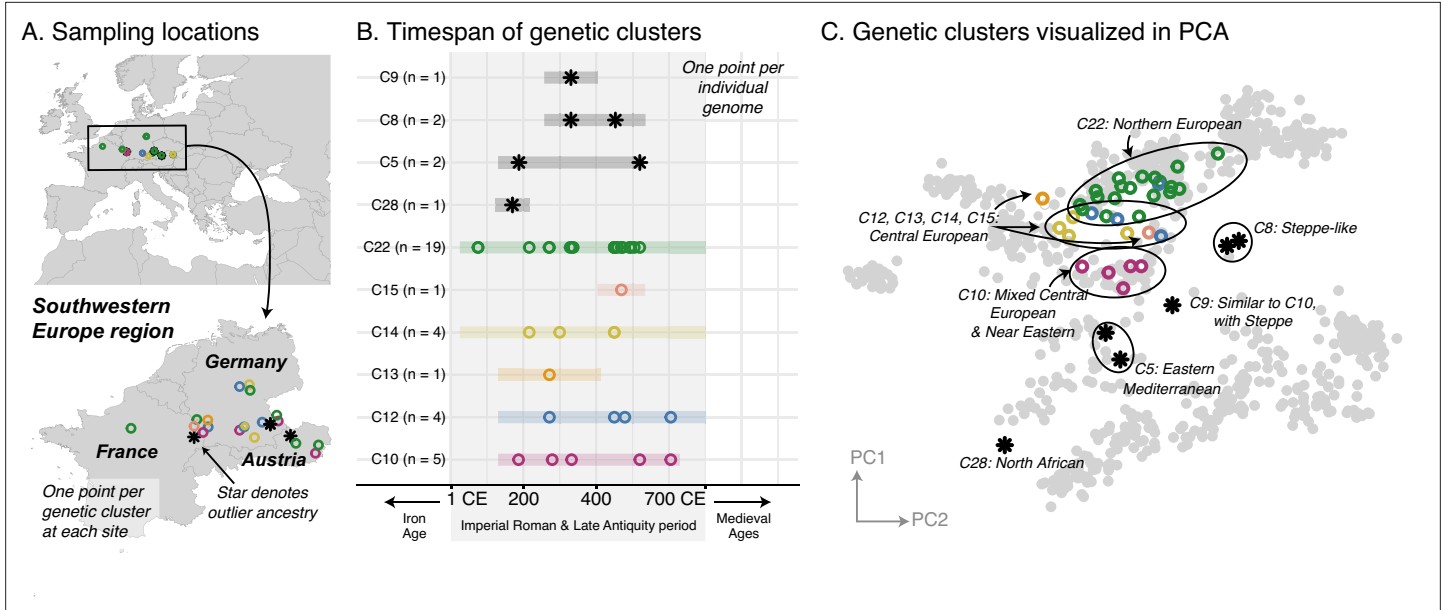

**Figure 4.** Western Europe: heterogeneous Imperial Roman and Late Antiquity period population. (**A**) Sampling locations of genetic clusters are represented by a single point per location. Outlier ancestries are black stars, all others are open circles colored by genetic cluster. (**B**) Colored bars span the minimum and maximum of the date ranges of samples (95% confidence interval from radiocarbon dating or archaeological range). Points are the mean of an individual's date range. (**C**) Projections of the ancient genomes onto a PCA of present-day genomes (gray points). Population labels for the PCA reference space are shown in **Figure 2C**. Present-day genomes from Southeastern Europe are shown with dark gray open circles.

European individuals during the Imperial Roman & Late Antiquity period also exhibit high heterogeneity, on par with that of contemporaneous Italy (**Figures 3 and 4**).

Furthermore, these ancestries are often shared across regions. In Southeastern Europe, a core group of individuals have ancestry similar to that of present-day and contemporaneous Central Europeans (C6), while other clusters have ancestry similar to that of Northern Europeans (C7) and Eastern Mediterraneans (C8) (**Figure 3C**). These ancestry groups are found in contemporaneous Italy and Western Europe as well (**Figure 4C**, **Figure 3—figure supplement 1**). We also observe individuals of eastern nomadic ancestry, similar to that of Sarmatian individuals previously reported, in both Western Europe (C8, n=2) and Southeastern Europe (C2, n=2).

Overall, we see remarkable local genetic heterogeneity as well as cross-regional similarities which point to common ancestry sources and, on a broader scale, demographic events affecting different regions in similar ways.

## At least 7-11% of historical individuals are ancestry outliers

The high regional genetic heterogeneity with long range, cross-regional similarities suggests historical populations were highly mobile. We therefore sought to quantify the amount of movement during the historical period by estimating the proportion of individuals who are ancestry outliers with respect to all individuals found in the same region. We considered an individual an outlier if they belonged to an ancestry cluster that is underrepresented (consisting of fewer than 5% of individuals in a region or at most two individuals) within their sampling region from the Bronze Age up to present-day. To focus on first-generation migrants as well as long-range movements, we further identified outlier individuals who can be modeled as 100% (i.e. 'one-component model') of a majority ancestry cluster found in a different region.

In total, we identified 11% of individuals as outliers, and could connect 7% of individuals to a putative source in a different region (**Figure 5A**). Based on the regions where these outliers and their sources originated, we created a network to illustrate their movements (**Figure 5B**). This network reveals the interconnectedness of Europe and the Mediterranean during the historical period. For example, as discussed above, the Armenian population is quite homogeneous (**Figure 2**). Unsurprisingly, no outliers were found within Armenia; however, we found outlier individuals in the Levant

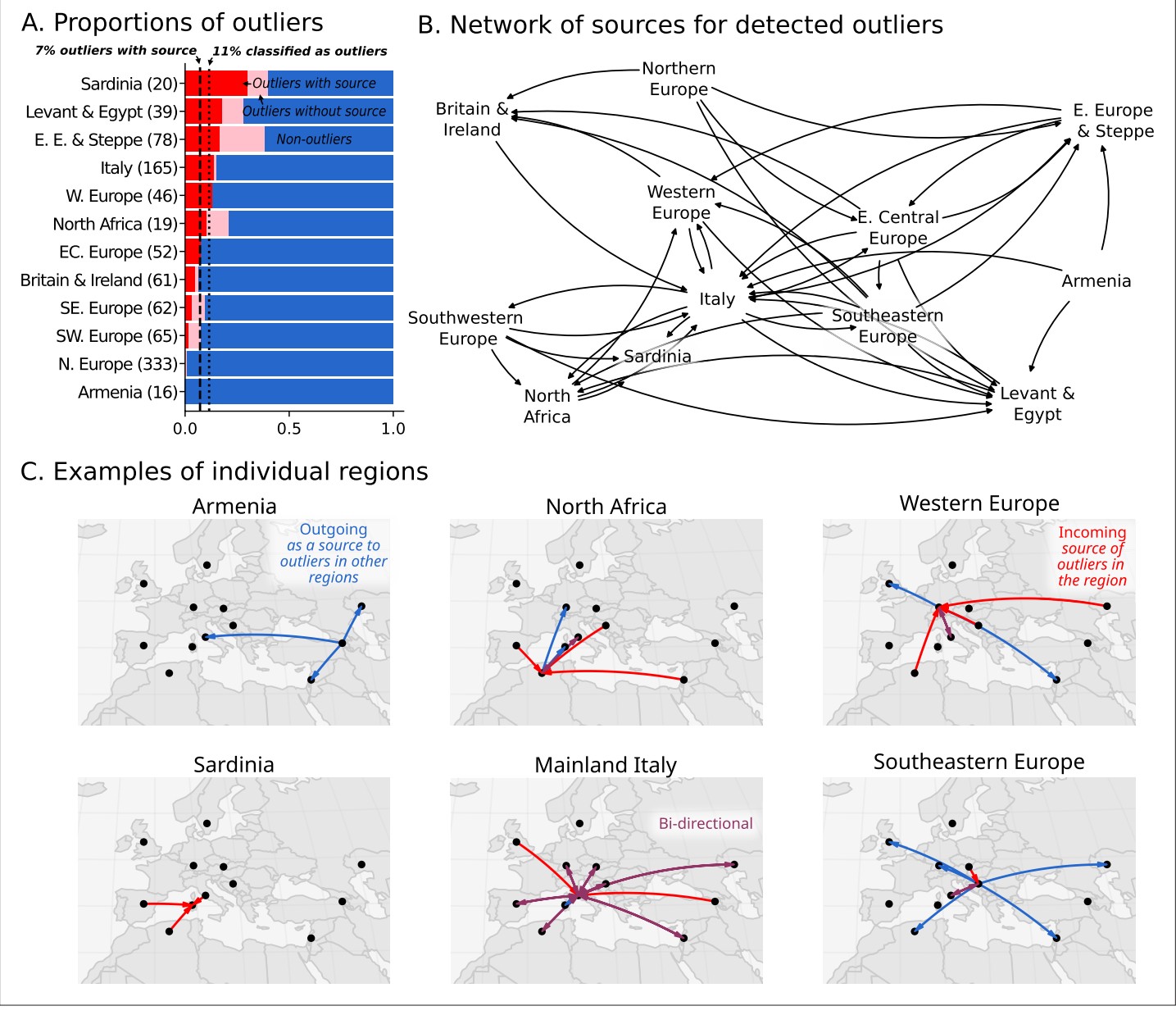

**Figure 5.** Ancestry outliers and their potential sources. (**A**) The proportions of outliers in each region were determined by individual pairwise qpAdm modeling followed by clustering. (**B**) Sources were inferred by one component qpAdm modeling of resulting clusters with all genetic clusters in the dataset. In the network visualizations, nodes are regions and directed edges are drawn from sources to outliers (i.e. potential migrants). The full network of source to outlier is shown. (**C**) Examples of individual regions are shown in greater detail.

The online version of this article includes the following figure supplement(s) for figure 5:

**Figure supplement 1.** Lack of sex-bias amongst outliers with valid qpAdm sources.

**Figure supplement 2.** Distances of outliers to their candidate sources.

**Figure supplement 3.** Example routes and travel times across the Roman Empire.

and Italy who can be putatively traced back to Armenia according to their ancestry (***Figure 5C***; blue outgoing arrows from Armenia). In contrast, the heterogeneous population in Italy connects it to many other regions, with bi-directional movement in most cases. In North Africa, outliers found in Iron Age Tunisia (***Moots et al., 2022***) indicate movements from many regions in Europe, and North African-like outliers were found in Italy and Austria (Western Europe). North African ancestry in Italy is supported by a single previously reported individual from the Imperial Roman period (R132; ***Antonio***

*et al., 2019*). Similar North African ancestry in Western Europe is supported by a single individual, R10667, from Wels, Austria, a site located on the frontier of the Roman Empire (C28 in *Figure 4*). This individual from Austria can be modeled using Canary Islander individuals from the Medieval Ages or an Iron Age outlier (distinguished by having more sub-Saharan ancestry) from Kerkouane, a Punic city near Carthage in modern-day Tunisia.

The 7% estimate for outliers with source should be considered conservative for the proportion of 'non-local' individuals. There are several cases where a cluster comprises more than 5% of the individuals in the region, but are clearly of a different ancestry than the majority and seem to be transient (only found in a single sub-period of the historical period). For example, in Southeastern Europe (*Figure 3B*), Imperial Roman & Late Antiquity individuals in C8 are (1) of distant ancestry (Near Eastern) and (2) not found in previous or subsequent time periods. However, since there are five individuals in this cluster, it does not meet our strict criteria for outlier consideration. Additionally, many clusters of underrepresented ancestry cannot be modeled as one-component models because they are recently admixed (i.e. require two or more ancestry components) or of ancestry not sampled elsewhere. Thus, we expect the actual proportion of individuals involved in long distance movements to be higher than reported here.

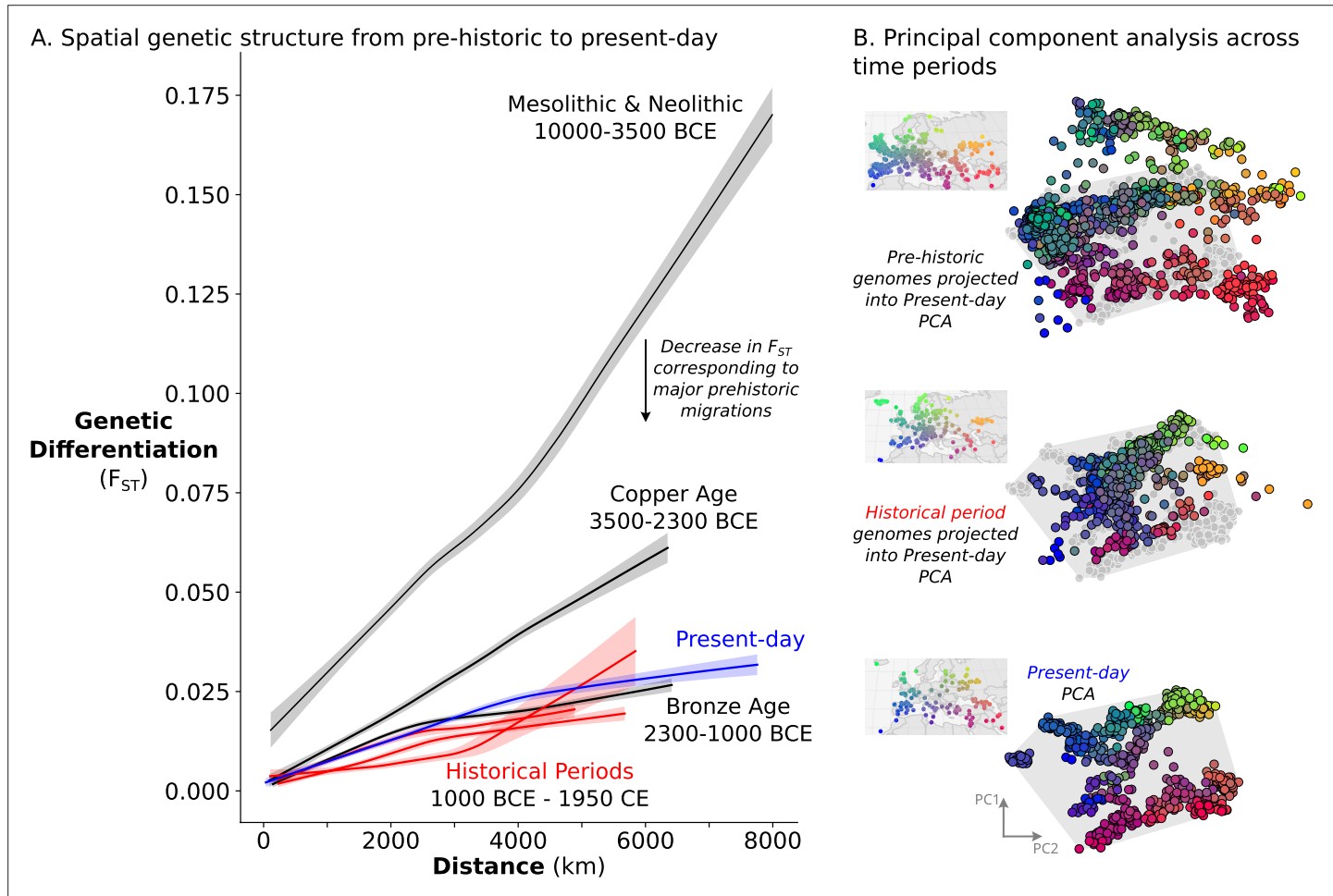

**Figure 6.** Relatively stable population structure from Bronze Age to present-day. (**A**) Overall genetic differentiation between populations (measured by $F_{ST}$) and its relationship to geographical distance (spatial structure) is similar from Bronze Age onward. Confidence intervals were calculated through a bootstrap procedure, using 200 bootstrap replicates. (**B**) In PC space, each genome is represented by a point, colored based on their origin (for present-day individuals) or sampling location (for historical samples). The PC space is established by present-day samples (bottom), onto which either historical period (middle) or prehistoric genomes (top) were projected. For projections, the present-day samples are shown in gray, and their extent is visualized by a gray polygon.

## Spatial population structure is relatively stable in the last 3,000 years

The remarkable amount of heterogeneity and mobility in the historical period leads to the question of what impact this might have had on population structure over time. To investigate this, we sought to quantify the overall change in population structure across time, from prehistoric to present-day. To assess the spatial structure of population differentiation, we calculated $F_{ST}$ across groups of individuals on a sliding spatial grid in each time period and related it to their mean geographic distance. In each time period, we recovered the classical pattern of isolation-by-distance (*Figure 6A*), where individuals closer in geographic space are also more similar genetically. Across time periods, we see a large decrease in overall $F_{ST}$ from the Mesolithic & Neolithic periods to the Bronze Age (approximately 10,000–2300 BCE), coinciding with the major prehistoric migrations (*Haak et al., 2015*; *Lazaridis et al., 2014*). From the Bronze Age onward, however, $F_{ST}$ does not decrease further with time, indicating that the level of genetic differentiation across space is relatively stable from the Bronze Age to present-day.

To assess not only the amount, but also the structure of geographic population differentiation, we compared the 'genetic maps' of historical period and present-day genomes. To construct these 'maps', we performed principal component analysis on 829 present-day European and Mediterranean genomes sampled across geographical space (*Figure 6B*, bottom) and projected historical period genomes onto the same PC space. Echoing close correspondence between genetic structure and geographic space in present-day Europeans (*Novembre et al., 2008*), we recovered similar spatial structure for historical samples as well, although noisier due to a narrower sampling distribution and higher local genetic heterogeneity (*Figure 6B*, middle). The similarity in structure between present-day and historical period is especially striking in comparison to a projection of prehistoric genomes, which shows much weaker correspondence to the present-day PCA as well as to geographic space (*Figure 6B*, top). Together, our analyses indicate that European and Mediterranean population structure has been relatively stable over the last 3000 years.

This raises the question: is it surprising for stable population structure to be maintained in the presence of ~7–11% long-range migration? To address this, we simulated Wright-Fisher populations evolving neutrally in continuous space. In these simulations, spatial population structure is established through local mate choice and limited dispersal, which we calibrated to approximately match the spatial differentiation observed in historical-period Europe (*Figure 6A*, *Figure 7A* and *Figure 7—figure supplement 1*, maximum $F_{ST}$ of ~0.03). We then allowed a proportion of the population to disperse longer distances, empirically matching the migration distances we observed in the data during the historical period (*Figure 7—figure supplement 2*). Even with long-range dispersal as low as 4%, we observe decreasing $F_{ST}$ over 120 generations (~3000 years with a generation time of 25 years) as individuals become less differentiated genetically across space (*Figure 7B*). At 8%, $F_{ST}$ decreases dramatically within 120 generations as spatial structure collapses to the point that it is hardly detectable in the first two principal components (*Figure 7C*). These simulations indicate that under a basic spatial population genetics model we would expect structure to collapse by present-day given the levels of movement we observe.

## Discussion

In summary, we observed largely stable spatial population structure across western Eurasia and high mobility of people evidenced by local genetic heterogeneity and cross-regional connections. These two observations are seemingly incompatible with each other under standard population genetics assumptions.

A possible explanation for this apparent paradox is that our simulations did not capture some key features of human behavior and population dynamics. In the simulated populations, migration implies both movement and reproduction with local random mate choice. However, in real human populations migration can be more complex: people do not necessarily reproduce where they migrate, and reproduction is not necessarily random. We hypothesize that in the historical period there was an increasing decoupling of movement and reproduction, compared to prehistoric times. For the spread of Farmer and Steppe ancestry, we know that these prehistoric migrations would take hundreds of years to traverse the continent (*Allentoft et al., 2015*; *Haak et al., 2015*; *Lazaridis et al., 2016*). In contrast, in the historical period, there were dense travel networks of roads and waterways as well

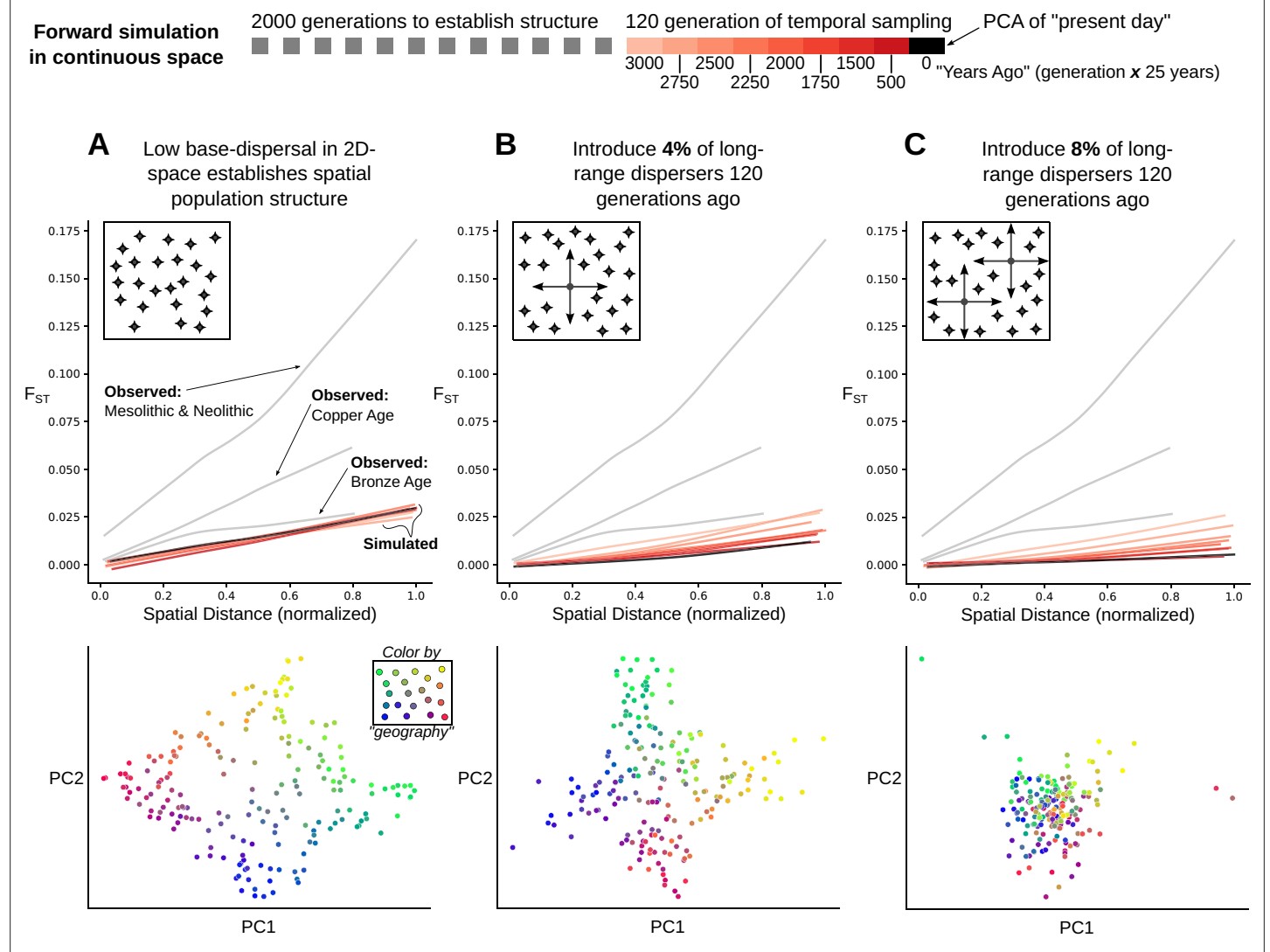

**Figure 7.** Simulation of population structure with and without long-range dispersal. (**A**) A base model of spatial structure is established by calibrating per-generation dispersal rate to generate a maximum $F_{ST}$ of ~0.03 across the maximal spatial distance, and visualized using PCA. In addition to this base dispersal, either 4% (**B**) or 8% (**C**) of individuals disperse longer distances, and the effect is tracked by analyzing spatial $F_{ST}$ through time, as well as PCA after 120 generations of long-range dispersal.

The online version of this article includes the following figure supplement(s) for figure 7:

**Figure supplement 1.** A *sigmaDisp - N* parameter pair was chosen to closely approximate the observed $F_{ST}max$ of ~0.03 using grid search across a range of parameter pairs.

**Figure supplement 2.** A *sigmaDispLR* parameter was chosen to qualitatively resemble long-range dispersal distances observed in the data, by comparing the distribution of distances under long-range dispersal (*outliers*) to randomly chosen distances given the spatial distribution of samples.

as clear incentives for cross-Mediterranean and cross-continental movement (*Abulafia, 2011*; *Beard, 2015*; *Broodbank, 2013*; *Symonds, 2017*). This enabled people to travel cross-continental distances on the order of weeks or months, well within their lifetimes (*Figure 5—figure supplements 2 and 3*; *Scheidel, 2015*).

The Roman Empire is particularly important in understanding how transient mobility could become a unique hallmark of this period. During the expansion of the Empire, existing and new cities quickly expanded as hubs for trade and labor. Urban-military complexes emerged along the frontier as military forces established themselves and drew in local communities which sought protection or economic benefit (*Verhagen et al., 2019*). To support these rapidly growing economic city-centers, human capital beyond the local population was necessary, thus drawing in people from far away places either

freely or forcibly (e.g. slavery, military). According to a longstanding historical hypothesis, the Urban Graveyard Effect, the influx of migrants in city-centers disproportionately contributed to the death rate over the birth rate; a process which would contribute to observing individuals as 'transient' migrants (*Tacoma, 2016*). Long-range, transient migration, combined with the Roman Empire's highly efficient travel networks (*Scheidel et al., 2007*; *Oleson, 2009*; *Scheidel, 2015*) may explain the genetically heterogeneous populations, especially along the frontier regions (e.g. Serbia, Croatia, and Austria).

With transient mobility as the main contributor to the observed heterogeneity, it remains unclear what additional demographic processes contributed to the maintenance of spatial genetic structure. The collapse of the Empire involved a loss of urban-military complexes and depopulation of cities, followed by ruralization (*Burgess, 2007*; *Dey, 2015*; *Roymans et al., 2020*). Without the Empire incentivising trade and movement, there may have been little motive for individuals to remain in these suddenly remote regions.

If this hypothesis is true, we would expect a reduction in local genetic heterogeneity after the collapse of the Empire. Unfortunately, we do not have this period sampled densely enough to assess this comprehensively. The lack of samples is further amplified by the fact that ancient DNA comes from archaeological excavations, which tend to be enriched in urban areas; a stone mausoleum in the city center, for example, will produce more surface scatter than a wood farmhouse, making urban areas more likely for excavation (*Bowes, 2011*). This makes it difficult to comprehensively address differences in rural versus urban demography. Collecting more genetic data from both urban and rural contexts across the historical period will be a valuable future step in understanding how spatial population structure was maintained. Furthermore, it could elucidate the role of other historical events and peoples, such as the Franks, Lombards, Visigoths, and Huns, during the Migration Period.

Based on genetic analyses and the rich historical record, we hypothesize that both the loss of transient migrants which contributed to population heterogeneity, as well as repopulation by less heterogeneous, but temporally stable, local populations could have helped maintain overall stability of genetic structure from the Iron Age to present-day. This work highlights the utility of ancient DNA in revealing complex population dynamics through direct genetic observations through time and the importance of integrating historical contexts to understand these complexities.

## Materials and methods
### Sample collection and archaeological sites

The archaeological context for ancient individuals reported in this study is detailed in *Supplementary file 1*. Site descriptions were written by the contributing archaeologists. Descriptions of individual-level burials are included where possible.

Sampling was performed to maximize coverage across Europe and the Mediterranean, as opposed to detailed site level sampling. We particularly focused on regions where there was no published genomic data for the Imperial Roman and Late Antiquity Period at the time of sampling (e.g. France, Austria, the Balkans, Armenia, and North Africa).

Although we aimed to collect samples in the Imperial Roman and Late Antiquity period (approximately 1 CE-700 CE), some samples fall outside of this period due to limited sample availability and/or lack of date specificity at the time of sampling (prior to radiocarbon dating).

### Date determination for individuals and time periods

Determination of time periods and boundaries were influenced by (1) the wide geographic range represented in the newly reported and published data (including historical changes in those regions), (2) the amount of data available in the proposed time periods, (3) the amount of genetic change observed during a time interval, and (4) the types of temporal comparisons made in the analysis.

- Mesolithic and Neolithic 10000 BCE - 3500 BCE
- Copper Age 3500 BCE - 2300 BCE
- Bronze Age 2300 BCE - 1000 BCE
- Iron Age 1000 BCE - 1 CE
- Imperial Rome & Late Antiquity 1 CE - 700 CE
- Medieval Ages & Early Modern 700 CE - 1900 CE
- Present-day 1900 CE - onward

Dates for newly reported study individuals were determined through a combination of radiocarbon dating and archaeologically inferred dates. Sample groups were created based on the finest grouping using a composite of criteria: site, projection in PCA, and archaeologically inferred date or period. We aimed to radiocarbon date at least one sample from each of these groups, summing to a total of 126 samples. For each sample, one gram of petrous bone was sent for radiocarbon dating by Accelerator Mass Spectrometry (AMS) at the Keck Carbon Cycle AMS Facility at the University of California, Irvine. Resulting dates were calibrated using https://c14.arch.ox.ac.uk/oxcal.html.

For samples not directly dated, we assigned the range of directly dated samples within their sample group (n=49), or the archaeological date where the former was not available (n=29). Note that one individual (R3477, R3476) had two radiocarbon dates since two samples (a tooth and petrous) were dated and sequenced before they were determined to be from the same individual based on genetic information. Another individual (R9818, R9823) had two radiocarbon dates because both left and right petrosals were sampled, but the sequence data for one turned out to be contaminated. In both cases of identical samples the radiocarbon date ranges were almost entirely overlapping.

In the study we use a single date estimate (for both newly reported and published samples) which is the midpoint of the 95% confidence interval when using AMS dates, and the average of the lower and upper bound inference dates when using archaeological context for dating. The dating approach used for each sample is included in files *Supplementary file 1* (archaeological context) and *Supplementary file 2* (sample metadata). The full AMS and calibration results are reported in *Supplementary file 3*.

## DNA extraction, library preparation, and sequencing

The 204 ancient genomes reported in this study, 26 of which were recently reported in *Moots et al., 2022*, represent a subset of samples screened from 53 archaeological sites across 18 countries. We isolated and finely ground the cochlear regions of the petrous bones in dedicated clean room facilities at the University of Vienna following the protocols described in *Pinhasi et al., 2019*; *Pinhasi et al., 2015*. Using 50 mg of bone powder, DNA was extracted by 18 hr incubation of the powder in a solution of Proteinase-K and EDTA. DNA was eluted in 50 µl 10 mM Tris-HCl, 1 mM EDTA, 0.05% Tween-20, pH 8.0 as in *Dabney et al., 2013*; *Rohland and Hofreiter, 2007*. 12.5–25 µL of DNA extract was used to prepare partial uracil–DNA–glycosylase (UDG) double stranded libraries as described in *Rohland et al., 2015*. After a partial (30 minute) UDG treatment, library preparation followed a modified version of the *Meyer and Kircher, 2010* protocol (*Meyer and Kircher, 2010*): the initial DNA fragmentation step was not required and MinElute PCR purification kits (Qiagen) were used for all library clean-up steps. Libraries were measured on a qPCR to determine the ideal cycle number to avoid over or under-amplification. 10-20 uL of library was then double indexed using Agilent PfuTurbo Cx HotStart DNA Polymerase with conditions: 95 °C for 5 min followed by the qPCR-determined cycles of 95 °C for 15 s, 60 °C for 30 s and 72 °C for 30 s with a final elongation at 72 °C for 5 min. After indexing, the libraries were purified using the MinElute system (Qiagen) and eluted in 25 µL of 1 mM EDTA, 0.05% Tween-20. Libraries were screened based on Qubit concentration and visual validation of Bioanalyzer peaks for an initial low coverage (NextSeq or Novaseq) screening run.

## Processing sequence data and sample screening

Newly reported samples were initially sequenced to low coverage on MiSeq or NextSeq for screening. Following demultiplexing of the sequencing libraries, reads were trimmed, aligned, filtered for quality, and deduplicated. The following sequence data processing pipeline was applied to both screening and full sequencing runs for new data.

Adapters were removed from sequence reads using Cutadapt (v1.14) (*Martin, 2011*). Then, for each sample, reads were processed further (a) with the 2 base pairs at either end of the reads trimmed off and (b) without trimming. Since partial UDG treatment was performed on the libraries, a damage signature consisting of elevated C>T transitions on the 5' end and G>A transitions on the 3' end should remain at the ends of reads. Therefore, analyzing untrimmed, aligned reads would allow us to assess the amount of the ancient DNA damage signature present in a sample, and to use this as a criteria for authenticating that the sampled DNA is ancient. Other than the variable trimming parameter for the ends of the reads, all other parameters remained the same for both screening and high coverage sequencing data.

Following variable trimming, reads were filtered for minimum length of 30, then aligned to hg19 using bwa (0.7.15-r1140; *Li and Durbin, 2009*), with seed length disabled (-l 350). For each sample, aligned reads were sorted by coordinate using Picard's SortSam (version 2.9.0–1-gf5b9f50-SNAPSHOT) and read groups were added using Picard's AddOrReplaceReadGroups (version 2.9.0–1-gf5b9f50-SNAPSHOT) (http://broadinstitute.github.io/picard/). Reads with mapping quality <25 (including unaligned reads) were filtered out. For higher coverage sequencing runs, this process was parallelized by splitting raw fastq files and merging after alignment, sorting, and quality filtering. Duplicates were removed using samtools rmdup (http://www.htslib.org/doc/samtools.html). Genome-wide and chromosomal coverage were assessed using depth-cover (version 1.0.3, https://github.com/jalvz/depth-cover; *Alvarez, 2017*).

Samples were screened and selected using the following criteria: (1)>20% reads aligned to the hg19 build of the human genome; (2) a C>T mismatch rate at the 5'-end and G>A at the 3'-end of the sequencing read of 4% or above (characterized with mapDamage v2.0.8) (*Jónsson et al., 2013*); (3) library complexity estimates indicating that a minimum coverage of 0.5 x would be achievable with further sequencing, (4) with a contamination level ≤ 5%.

Contamination rates were estimated with three methods: (1) damage pattern and polymorphism in mitochondrial DNA with Schmutzi (*Renaud et al., 2015*), (2) atypical ratios of coverages of X and Y chromosomes to autosomes calculated with ANGSD (*Korneliussen et al., 2014*), and (3) for male samples, high heterozygosity on non-pseudo-autosomal region of the X chromosome (chrX:5000000–154900000 in hg19) with the 'contamination' tool in ANGSD (*Korneliussen et al., 2014*). If the contamination estimate for any of these three methods was above 5%, we considered the sample contaminated and excluded it from downstream analysis (n=9).

For passing samples, processed data from all sequencing runs were merged into a single BAM file. The 204 new samples that passed quality filters have a median genome-wide depth of 0.92 x (0.16x to 2.38x).

## Calling pseudohaploid genotypes

Pseudohaploid genotypes for study samples were called by randomly choosing one allele from each site where there was read coverage, following the approach and software provided by Stephan Schiffels (https://github.com/stschiff/sequenceTools; *Schiffels, 2022*). Variants were called for the 1240 k SNP panel, which is commonly used for capture-based sequencing of ancient samples (*Mathieson et al., 2015*). For the newly reported samples, a median of 685,058 SNPs (167,000–1,029,345) were covered per sample. Data was output in eigenstrat format. This pipeline was also used to call genotypes for two published ancient DNA datasets which at the time were only available in BAM (sequence read) format (*Clemente et al., 2021*; *Žegarac et al., 2021*).

## Combining new genotypes with ancient and present-day published data

Newly processed pseudohaploid data was merged with several datasets. Most of the published data was retrieved from the Allen Ancient Data Resource (AADR) v44.3 (January 2021) (*Allen Ancient DNA Resource, 2021*; *Mallick et al., 2023*): a compilation of pseudohaploid and diploid genotypes for 5,225 ancient and 3,720 present-day individuals (1000 *Auton et al., 2015*; *Agranat-Tamir et al., 2020*; *Allentoft et al., 2015*; *Amorim et al., 2018*; *Antonio et al., 2019*; *Bergström et al., 2020*; *Biagini et al., 2019*; *Brace et al., 2019*; *Broushaki et al., 2016*; *Brunel et al., 2020*; *Cassidy et al., 2020*; *Cassidy et al., 2016*; *Damgaard et al., 2018*; *de Barros Damgaard et al., 2018*; *Ebenesersdóttir et al., 2018*; *Feldman et al., 2019a*; *Feldman et al., 2019b*; *Fernandes et al., 2020*; *Fernandes et al., 2018*; *Fregel et al., 2018*; *Fu et al., 2016*; *Furtwängler et al., 2020*; *Gamba et al., 2014*; *Gokhman et al., 2020*; *González-Fortes et al., 2019*; *González-Fortes et al., 2017*; *Günther et al., 2018*; *Günther et al., 2015*; *Haber et al., 2020*; *Haber et al., 2019*; *Haber et al., 2017*; *Harney et al., 2018*; *Broushaki et al., 2016*; *Järve et al., 2019*; *Jeong et al., 2019*; *Jones et al., 2017*; *Jones et al., 2015*; *Keller et al., 2012*; *Kılınç et al., 2016*; *Krzewińska et al., 2018b*; *Krzewińska et al., 2018a*; *Lamnidis et al., 2018*; *Lazaridis et al., 2017*; *Lazaridis et al., 2016*; *Lazaridis et al., 2014*; *Linderholm et al., 2020*; *Lipson et al., 2017*; *Mallick et al., 2016*; *Malmström et al., 2019*; *Morrison et al., 2020*; *Margaryan et al., 2020*; *Martiniano et al., 2017*; *Martiniano et al., 2016*; *Mathieson et al., 2018*; *Mathieson et al., 2015*; *Mittnik et al., 2019*; *Mittnik et al., 2018*; *Narasimhan et al.,*

*2019*; *Nikitin et al., 2019*; *Olalde et al., 2019*; *Olalde et al., 2018*; *Olalde et al., 2015*; *Olalde et al., 2014*; *Omrak et al., 2016*; *O'Sullivan et al., 2018*; *Patterson et al., 2012*; *Prüfer et al., 2017*; *Rivollat et al., 2020*; *Rodríguez-Varela et al., 2017*; *Saag et al., 2019*; *Saag et al., 2017*; *Sánchez-Quinto et al., 2019*; *Schiffels et al., 2016*; *Schroeder et al., 2019*; *Schuenemann et al., 2017*; *Sikora et al., 2017*; *Skoglund et al., 2014*; *Skourtanioti et al., 2020*; *Unterländer et al., 2017*; *Valdiosera et al., 2018*; *van den Brink et al., 2017*; *Veeramah et al., 2018*; *Villalba-Mouco et al., 2019*; *Wang et al., 2019*; *Zalloua et al., 2018*). We also included relevant genetic data made available by authors that were not in the AADR: present-day genomes from the Balkans (*Kovacevic et al., 2014*), present-day genomes from 4 Poles, 3 Germans, and 2 Moldavians (*Pagani et al., 2016*), and Bronze Age Italian genomes (*Saupe et al., 2021*). Pseudohaploid genotypes for published Bronze Age Aegean genomes (*Clemente et al., 2021*) and Bronze Age Serbian genomes (*Žegarac et al., 2021*) were generated from BAM files using our pipeline. All published genomes were filtered for contamination based on reported contamination levels in the original study and SNP coverage based on the genomic data. All published samples that contributed to this study are listed in *Supplementary file 4*. To ensure maximum overlap with present-day and ancient samples in analyses, the merged dataset was subset to SNPs in the Human Origin Panel array, resulting in a total of 481,259 SNPs. For PCA and *qpAdm* modeling, SNPs that are transitions at CpG sites (n=76,678) were excluded since they may have arisen from DNA damage as opposed to true genetic variation.

## Principal component analysis (PCA)
### Setting up the principal component analysis
Principal component analysis was performed on genotypes from present-day and Mediterranean individuals using smartpca v16000 (https://github.com/chrchang/eigensoft/blob/master/POPGEN/README; *Chang, 2013*). The following parameters were used: 5 outlier iterations (numoutlieriter), 10 principal components along which to remove outliers (numoutlierevec), altnormstyle set to NO, with least squares projection turned on (lsqproject set to YES). To calculate principal components only using present-day individuals, a file (poplistname) was provided with the population names of present-day individuals, randomly subsampled per population. After outlier removal (which removed 55 samples), 829 individuals and 480,712 SNPs were used in the initial analysis. All individuals (non 'reference' present-day genomes, and all of the ancient individuals) whose population was not listed in the poplistname file were projected onto the calculated principal components. In the paper, we refer to the individuals used in the calculation of principal components as belonging to the 'reference PCA space'. These 'reference' genomes were used to calculate the PCs because (1) they represent a wide range of present-day variation and (2) the genotypes tend to be of high quality.

### Visual representation of PCA
In the figures, present-day genomes used in the reference space are generally colored gray in order to illustrate the background space of genetic variation. To reduce visual clutter and emphasize the ancient genomes, these present-day 'reference' genomes are typically unlabeled. Labels for these populations are shown in *Figure 2—figure supplement 1*.

## Calculation of $F_{ST}$
To assess the extent of genetic differentiation across geographic space within a time period, we calculated the Fixation index ($F_{ST}$) between groups of individuals on a sliding spatial grid. Each grid cell measured 10 degrees longitude by 10 degrees latitude, and was slid by 1 degree in both directions (north and east) nine times to build a total of 10 spatial grids. For each of these grids, pairwise $F_{ST}$ was calculated between all populated 10-by-10 grid cells using Hudson's estimator, correcting for unequal sample size (*Bhatia et al., 2013*). In addition to $F_{ST}$, we also calculated the average geographic distance (in kilometers) between all individuals across pairs of grid cells to assess how spatial distance relates to genetic differentiation. To visualize this relationship, we used lowess smoothing as implemented in python's *statsmodels* package (*statsmodels.api.nonparametric.lowess*, v. 0.12.2). To infer confidence intervals for the lowess smoothing estimates, we devised a spatial bootstrapping procedure. Our bootstrap approach samples pairs of grid cells in a way that always samples all overlapping cells or none of them, so individuals are either fully included in a bootstrap replicate or not at all. This

prevents double-counting individuals since they contribute to several comparisons across space due to the sliding grid.

## Modeling ancestry and identifying outliers using *qpAdm*

We used the *qpAdm* tool of *admixtools 2.0* to build a workflow that:

1. Identifies similar individuals within a region
2. Groups these into regional clusters
3. Compares these clusters both within and across regions

In this workflow, we heavily utilize what we call one-component models, where we test whether two individuals or clusters of individuals form a clade relative to a chosen set of reference populations (which we define below). To clarify what we mean by one-component model, assume that we have two focal individuals *i1* and *i2*, as well as a set of reference populations. Using the terminology of *admixtools 2* as well as (**Harney et al., 2021**), the following four tests are equivalent with respect to the resulting p-value:

- *qpWave(left = c(i1, i2), right = ref)*
- *qpAdm(left = c(i1, i2), target = NULL, right = ref)*
- *qpAdm(left = i1, target = i2, right = ref)*
- *qpAdm(left = i2, target = i1, right = ref)*

where we use the *R-style* notation of *c(i1, i2)* to denote a vector consisting of *i1* and *i2*. In all four cases, we are testing against the null hypothesis that the two individuals *do* form a clade, with low p-values indicating a rejection of that hypothesis. We will call any implementation of this test a 'one-component *qpAdm* model', but the equivalence stated above shows that our one-component models are equivalent to what was called a 'qpWave analysis' by **Fernandes et al., 2020**.

For our reference populations, we chose a set similar to those previously used to model Eurasian historical genomes (**Fernandes et al., 2020**). However, we added two Asian populations (Laos_Hoabinhian and Onge) based on evidence of gene flow from further east in a subset of our data. The purpose of a set of reference populations in the *qpAdm* modeling setting is to represent components of ancestry which are differentially related to the focal individuals being tested (i.e. 'left', or 'sources' and 'target') in order to resolve differences in ancestry, but distally related enough to minimize the chance of recent gene flow between 'left' and 'right'. Our final set of references is:

Mbuti.DG (n=4), WHG (n=8), Russia_Ust_Ishim.DG (n=1), CHG (n=2), EHG (n=3), Iran_GanjDareh_N (n=8), Israel_Natufian_published (n=3), Jordan_PPNB (n=6), Laos_Hoabinhian (n=1), Russia_EBA_Yamnaya_Samara (n=9), Onge (n=6), Spain_ElMiron (n=1), Turkey_N_published (n=8), Russia_MA1_HG (n=1), Morocco_Iberomaurusian (n=6), Czech_Vestonice16 (n=1).

### Individual-based one-component models within regions

In the first step of our workflow, we perform one-component *qpAdm* tests between all pairs of individuals from the same region, from the Copper Age (inclusive) up to present-day (exclusive). As mentioned above, this approach tests against the null hypothesis that the two individuals *do* form a clade, with low p-values indicating a rejection of that hypothesis. Low *qpAdm* p-values thus suggest that the test individuals are not more closely related to each other than to one or more populations in the reference set. To convert the *qpAdm* p-value into a measure of dissimilarity (*d*), we calculate $d = -log_{10}(p\text{-}value)$, where a large value of *d* indicates a low p-value, and thus a rejection of the null hypothesis of the two individuals forming a clade.

To cluster individuals into groups of genetically similar individuals, we performed hierarchical clustering on the dissimilarity matrix constructed from all pairwise values of *d* within a region. Hierarchical clustering was performed using the UPGMA algorithm as implemented in python's *scipy.cluster.hierarchy* (v. 1.6.1). The hierarchical clustering was then split into flat clusters using a dissimilarity cutoff of *1.3*, which corresponds to a nominal p-value cutoff of *0.05*. Intermediate results from the pairwise analysis and clustering are shown for each region in Appendix 1.

## Identifying ancestry outliers and their potential sources

Once clusters within regions were identified as described above, we classified them into two groups based on size. Clusters consisting of less than 5% of the total population or no more than two individuals in the region (across time) were classified as outlier candidates, whose ancestry is under-represented in the region they were sampled. All other clusters were classified as majority clusters. Following this classification of clusters, we then split each cluster by time period (Copper Age, Bronze Age, Iron Age, Imperial Rome & Late Antiquity, Middle Ages and Early Modern), to end up with a *region_period_cluster* sub-classification. All downstream analyses were done using these *region_period* clusters.

For each cluster identified as an outlier candidate, we then tested all one-component models involving the candidate and a majority cluster *within* the same region. This was done to ensure that outlier candidates were truly distinct from majority ancestries identified in the region. If an outlier candidate could be connected to a majority cluster within its region through a valid one-component model (above a p-value threshold of *0.01*), it was removed from the outlier candidate list, as it represented a majority ancestry within the region.

For all outlier candidates that remained, we aimed to identify potential source ancestries that were majority ancestries in other regions. To do so, we tested all one-component models for a given outlier candidate and each of the majority clusters *across* all other regions. A majority cluster from a different region that had a valid one-component model with an outlier candidate was considered a potential source. To find the most likely source(s), we then subjected these potential source populations to model competition.

Model competition involves re-testing the model fit of each potential source after adding another potential source to the right group, first described in *Lazaridis et al., 2016*, and more recently detailed in *Harney et al., 2021*. The idea is that if a population in the right group has significantly more allele sharing with the target than the source, then the model will be rejected. (This is why right group populations are chosen to be distal, yet relevant, to target and source). We use this property to our advantage by rotating all n-1 valid sources for the target through the right group set, one at a time, for the same target and a source *x* (the one source that is not included in the rotation). If the previously valid source *x* is rejected when including another valid source *y* in the right group then we remove source *x* from the list of potential sources. Note that this does not make source *y* the best source, only a better one than *x*. Thus, this scheme only eliminates sub-optimal sources, rather than selecting a best source.

If there were still multiple valid sources following the model competition scheme described above, we prioritized a candidate source that is from the same time period as the target cluster, or from a previous period going backwards in time. If there were still multiple candidate sources, they were considered equivalent and all kept in downstream analyses.

Among the outliers identified, we did not find a significant sex bias compared to non-outliers. Overall, there are more males than females in the dataset. However, the proportions of males in non-outliers, outliers with source, and outliers without source do not differ significantly by a Chi-squared test (p-value = 0.4117, df = 2; *Figure 5—figure supplement 1*). When outliers (with and without source) are treated as one group, there is still no significant association with outlier status and sex (p-value = 0.633, df = 1).

## Admixture modeling with *qpAdm*

For targeted analyses of other clusters beyond just outlier candidates, for example to annotate *Figures 2–4*, we also used cluster-based *qpAdm*. In addition to one-component models as described above, we also used two-component models of admixture. These models test the hypothesis that a focal target cluster can be modeled as a two-way admixture of two sources (or 'left' populations). As above, a p-value below the threshold rejects this hypothesis, that is the proposed admixture model is not a good fit and a different model needs to be considered to disentangle the admixture scenario in question.

## Simulations

To assess how spatial population structure would be impacted by different modes of dispersal, we set up forward simulations in continuous 2D space using SLiM v. 3.6 (*Haller and Messer, 2019*). The aim of these simulations was to approximate the extent of spatial population structure we observe

by the beginning of the Iron Age in Western Eurasia, after the major prehistoric migrations had taken place. To achieve that, we decided not to attempt simulating the precise ancestry composition of populations in different regions at that time, but rather to simulate simply the extent of spatial structure as measured by the relationship of population differentiation ($F_{ST}$) and geographic distance. We chose the SLiM simulation framework to make use of its extensive feature set to simulate individuals in continuous space. We simulated diploid genomes made up of a single, $10^8$ bp long chromosome, with recombination rate and mutation rate set to $10^{-8}$. We used the default Wright-Fisher simulation mode, where a single population of constant size $N$ is simulated with non-overlapping populations, that is each generation is made up of offspring generated from the previous generation. Spatial structure is established by associating each individual with a continuous 2D coordinate (i.e. latitude and longitude), and by using these coordinates to govern three demographic processes: mate choice, competition, and dispersal. An overview of how these processes can be set up to interact in SLiM can be found in *Recipe 15.4* of SLiM v. 3.6 (see e.g. here: https://github.com/MesserLab/SLiM/tree/v3.6/SLiMgui/Recipes; *Haller, 2021*). Briefly, for mate choice, a Gaussian interaction function with *maxDistance = 0.1, maxStrength = 1.0, sigma = 0.02* is used to govern a *mateChoice* callback using the *strength* of that interaction function. For competition, another Gaussian interaction function with *maxDistance = 0.3, maxStrength = 3.0, sigma = 0.1* is used to calculate competition using the *totalNeighborStrength* vector of that interaction function to scale an individual's relative fitness as *1.1 - competition / N*. Finally, we establish local dispersal through a *modifyChild* callback, where a newly generated offspring's position is drawn from a Gaussian centered at the location of the maternal individual with standard deviation *sigmaDisp*.

We let this population evolve forward in time for 2000+120 generations during which the processes outlined above lead to spatial population structure, where individuals sampled closely together in 2D space are also more closely related genetically. We do not simulate mutations in SLiM, as this poses a major computational burden. Instead, we use tree sequence recording (*Haller et al., 2019*; *Kelleher et al., 2018*) to track the full genealogy of all individuals in the simulation which are either alive at the end of the simulation, or explicitly sampled through time using the *treeSeqRememberIndividuals* function of SLiM. While 2000 generations are enough to establish spatial structure under the parameters we consider, it is by far insufficient for all sampled individuals to fully coalesce. To accurately assess neutral variation however, we need all sampled individuals to have a common ancestor at some point in the past, as mutations may have arisen at any point leading back to this ultimate coalescence event. Therefore, we approximate the deep history of our population with a panmictic population simulated backwards in time using the coalescent with recombination as implemented in msprime (*Kelleher et al., 2016*). This process has been referred to as 'recapitation' (*Haller et al., 2019*), where an incomplete genealogy with multiple roots (from SLiM) is 'recapitated' using coalescent simulation backwards in time. This is made possible by using the tree sequence data structure to record and simulate genealogies in both SLiM and msprime.

Since our simulation is only concerned with how processes such as dispersal affect neutral variation across space and through time, we can use the 'recapitated' tree sequence to overlay mutations onto the full genealogy of all sampled individuals, also using msprime. The rationale here is that under neutrality, mutations will not affect the structure of the genealogy, so we can simulate the genealogy without mutations first, and overlay neutral mutations second, thereby greatly reducing computational burden.

We then extracted the resulting genotypes of all individuals from the tree sequence for downstream analysis. We only kept sites segregating in individuals at the end of the simulation ('present day'), and filtered for minor allele frequency of at least 0.01 across the entire dataset, to make downstream analysis of simulated genomes comparable to how the empirical data was ascertained and analyzed.

To assess the relationship between $F_{ST}$ and spatial distance, we split geographic space into a 10-by-10 grid and calculated all pairwise $F_{ST}$ between inhabited grid cells using Hudson's estimator with unequal sample size correction (*Bhatia et al., 2013*), as well as the average geographic (euclidean) distance between individuals across grid cells. We used this $F_{ST}$ analysis to calibrate the base dispersal *sigmaDisp* as well as the population size $N$, so that $F_{ST}$ at maximum distance ($F_{ST}$max) would approximately match the $F_{ST}$max we observed at the start of the historical period (~0.03). We used grid search with a range of *sigmaDisp* and $N$ values, and found the parameter pair $N=50,000$ & *sigmaDisp =*

*0.02* to qualitatively produce the closest match (*Figure 7—figure supplement 1*). We use this parameter set as our base model of population structure without long-range dispersal, where we allow spatial structure to establish over 2000 generations, and then observe the $F_{ST}$ - Distance relationship over the following 120 generations for a total of 2120 generations simulated in SLiM (*Figure 7A*).

Given this base model of spatial population structure, we can now start to introduce long-range dispersing individuals. We do this by allowing a specified fraction of individuals to use a higher *sigmaDispLR* than the *sigmaDisp* used by the rest of the population for the final 120 generations of the simulation, approximately matching the 3000 years since the beginning of the historical period assuming a generation time of 25 years. Since the long-range dispersal is also drawn from a Gaussian, the distribution of dispersal distances will have substantial overlap with the distances produced by base dispersal. To make the fraction of long-range dispersal accurately represent the fraction of individuals that actually disperse longer distances, we thus require a long-range dispersing individual to disperse to a location outside of the 99th percentile of density covered by the short-range base dispersal.

We aimed to choose a *sigmaDispLR* that approximately matches the empirical distribution of long-range dispersing individuals we observe in the analysis displayed in *Figure 4*. Since the euclidean distances in the simulation are on a different scale than the geodesic distances observed in the data, we aimed to match qualitatively the relationship of long-range dispersal distances to random distances that could be observed if two populated locations were drawn at random. We visually analyzed this relationship from the data, and then performed a search across a range of possible *sigmaDistLR* to find a qualitative match. This led us to choose a value of *sigmaDistLR = 0.20* (*Figure 7—figure supplement 2*).

Finally, we analyzed simulated 'present-day' genomes (i.e. after 2120 generations of SLiM) using PCA. We used the *sklearn.decomposition.PCA* module (scikit-learn v. 0.24.2) with the *svd_solver == 'arpack'* option to run non-probabilistic PCA to calculate the first 10 principal components. Similarly to how the empirical data was analyzed with *smartpca*, we also did 5 rounds of iterative outlier removal, removing individuals from the PCA that deviated by more than six standard deviations along any of the 10 principal components. The number of variants contributing to these PCA were 624,617, 625,669, and 626,052 for *Figure 7A, B and C* respectively, and thus comparable to the number of variants contributing to our data analysis.

## Ethics

This study follows ethics guidelines adopted by the ancient DNA field (*Alpaslan-Roodenberg et al., 2021*). A clear plan of research was laid out before the collection of samples, leading us to focus on sampling from under-sampled historical regions in Eurasia and therefore minimize unnecessary destruction of human remains. The research intent for these samples was clearly communicated to caretakers of the samples prior to collection. Local anthropologists, archaeologists, and museum directors from each geographic region were involved in the sample acquisition, extraction from skeletal material, and interpretation of genetic results. The genetic findings regarding individual samples from each region were communicated to local collaborators, all of whom were included as co-authors on the paper and were supportive of the final results. The involvement of our local collaborators was essential for the interpretation of the genetic results through their input on the historical and archaeological characterization of the specimens. We have supported our local collaborators with immediate access to the raw genetic data, and by communicating results in written and oral forums. Authorities responsible for all archaeological sites provided written documentation for their specimens to be included in this study through collaboration with the Pinasi Lab (Vienna, Austria).

## Acknowledgements

We thank Professor Walter Scheidel for helpful discussions and feedback on the historical context, and all members of the Pritchard and Pinhasi labs for their valuable input. We thank Pieter W Faber and the University of Chicago Genomics Facility for sequencing the samples reported here. We thank Benjamin Peter and an anonymous reviewer for their insightful and constructive reviews. This project was partially supported by a National Science Foundation Graduate Research Fellowship (MLA.), a grant from the National Institutes of Health RO1 HG011432 (CLW), the Austrian Science Fund (FWF) M3108-G (SS), and the Howard Hughes Medical Institute (JKP).

# Additional information

## Competing interests

Ziyue Gao: Reviewing editor, eLife. Maja Bilić: Affiliated with Palisada Ltd. The author has no financial interests to declare. Josip Burmaz, Dženi Los: Affiliated with Kaducej Ltd. The author has no financial interests to declare. Rene Masaryk: Affiliated with Skupina STIK. The author has no financial interests to declare. The other authors declare that no competing interests exist.

## Funding

| Funder | Grant reference number | Author |
|---|---|---|
| National Science Foundation | Graduate Research Fellowship | Margaret L Antonio |
| National Institutes of Health | HG011432 | Clemens L Weiß |
| Howard Hughes Medical Institute | | Jonathan K Pritchard |
| Austrian Science Fund | M3108-G | Susanna Sawyer |

The funders had no role in study design, data collection and interpretation, or the decision to submit the work for publication.

## Author contributions

Margaret L Antonio, Clemens L Weiß, Data curation, Formal analysis, Validation, Investigation, Visualization, Writing – original draft, Writing – review and editing; Ziyue Gao, Conceptualization, Formal analysis, Writing – original draft, Project administration, Writing – review and editing; Susanna Sawyer, Victoria Oberreiter, Olivia Cheronet, Brina Zagorc, Elisa Praxmarer, Kadir Toykan Özdoğan, Lea Demetz, Resources, Data curation; Hannah M Moots, Resources, Data curation, Writing – review and editing; Jeffrey P Spence, Data curation, Methodology; Pere Gelabert, Daniel Fernandes, Michaela Lucci, Timka Alihodžić, Selma Amrani, Pavel Avetisyan, Christèle Baillif-Ducros, Željka Bedić, Audrey Bertrand, Maja Bilić, Luca Bondioli, Paulina Borówka, Emmanuel Botte, Josip Burmaz, Domagoj Bužanić, Francesca Candilio, Mirna Cvetko, Daniela De Angelis, Ivan Drnić, Kristián Elschek, Mounir Fantar, Andrej Gaspari, Gabriella Gasperetti, Francesco Genchi, Snežana Golubović, Zuzana Hukeľová, Rimantas Jankauskas, Kristina Jelinčić Vučković, Gordana Jeremić, Iva Kaić, Kevin Kazek, Hamazasp Khachatryan, Anahit Khudaverdyan, Sylvia Kirchengast, Miomir Korać, Valérie Kozlowski, Mária Krošláková, Dora Kušan Špalj, Francesco La Pastina, Marie Laguardia, Sandra Legrand, Tino Leleković, Tamara Leskovar, Wiesław Lorkiewicz, Dženi Los, Ana Maria Silva, Rene Masaryk, Vinka Matijević, Yahia Mehdi Seddik Cherifi, Nicolas Meyer, Ilija Mikić, Nataša Miladinović-Radmilović, Branka Milošević Zakić, Lina Nacouzi, Magdalena Natuniewicz-Sekuła, Alessia Nava, Christine Neugebauer-Maresch, Jan Nováček, Anna Osterholtz, Julianne Paige, Lujana Paraman, Dominique Pieri, Karol Pieta, Stefan Pop-Lazić, Matej Ruttkay, Mirjana Sanader, Arkadiusz Sołtysiak, Alessandra Sperduti, Tijana Stankovic Pesterac, Maria Teschler-Nicola, Iwona Teul, Domagoj Tončinić, Julien Trapp, Dragana Vulović, Tomasz Waliszewski, Diethard Walter, Miloš Živanović, Mohamed el Mostefa Filah, Morana Čaušević-Bully, Mario Šlaus, Dušan Borić, Mario Novak, Resources; Alfredo Coppa, Resources, Supervision; Ron Pinhasi, Supervision, Funding acquisition; Jonathan K Pritchard, Conceptualization, Supervision, Funding acquisition, Methodology, Writing – review and editing

## Author ORCIDs

Margaret L Antonio http://orcid.org/0000-0003-1049-211X
Clemens L Weiß https://orcid.org/0000-0003-3321-3902
Ziyue Gao https://orcid.org/0000-0001-9244-0238
Victoria Oberreiter http://orcid.org/0000-0003-0766-3782
Jeffrey P Spence https://orcid.org/0000-0002-3199-1447
Brina Zagorc http://orcid.org/0000-0002-7685-1958
Christèle Baillif-Ducros http://orcid.org/0000-0003-3050-3082
Francesca Candilio http://orcid.org/0000-0002-4668-1361
Francesco Genchi http://orcid.org/0000-0002-7696-4207

Kristina Jelinčić Vučković http://orcid.org/0000-0002-1236-734X
Sylvia Kirchengast http://orcid.org/0000-0002-3220-7271
Wiesław Lorkiewicz http://orcid.org/0000-0003-0754-5161
Ana Maria Silva http://orcid.org/0000-0002-1912-6581
Jan Nováček http://orcid.org/0000-0003-2707-5347
Matej Ruttkay http://orcid.org/0000-0002-6441-9914
Arkadiusz Sołtysiak http://orcid.org/0000-0002-9040-5022
Alessandra Sperduti http://orcid.org/0000-0001-9338-5891
Tomasz Waliszewski http://orcid.org/0000-0002-5793-4600
Mario Šlaus http://orcid.org/0000-0002-4941-2212
Dušan Borić http://orcid.org/0000-0003-0166-627X
Mario Novak http://orcid.org/0000-0002-4567-8742
Alfredo Coppa http://orcid.org/0000-0002-7708-2484
Jonathan K Pritchard https://orcid.org/0000-0002-8828-5236

## Decision letter and Author response
Decision letter https://doi.org/10.7554/eLife.79714.sa1
Author response https://doi.org/10.7554/eLife.79714.sa2

---

# Additional files

## Supplementary files
• Supplementary file 1. Archaeological context for sampling locations.
• Supplementary file 2. Metadata for all newly reported individuals *Supplementary file 3*.
• Supplementary file 3. AMS and calibration results.
• Supplementary file 4. Published samples that contributed to this study.
• MDAR checklist

## Data availability
All sequence data newly generated for this study are available at the European Nucleotide Archive (ENA) database. Raw sequencing data is available under the accession number PRJEB53565. Sequences mapped to the human reference genome are available under accession number PRJEB53564. Sequences previously reported by *Moots et al., 2022* are available under accession number PRJEB49419. All published data used in this study is listed in *Supplementary file 4*, and can be retrieved from primary sources (see Materials and methods, 'Combining new genotypes with ancient and present-day published data') or from the Allen Ancient Data Resource (*Mallick et al., 2023*).

The following datasets were generated:

| Author(s) | Year | Dataset title | Dataset URL | Database and Identifier |
|---|---|---|---|---|
| Antonio, Weiß, Weiß, Sawyer et al. | 2022 | Historical Period Genomes from West Eurasia and the Mediterranean - Raw reads | https://www.ebi.ac.uk/ena/browser/view/PRJEB53565 | EBI European Nucleotide Archive, PRJEB53565 |
| Antonio, Weiß, Weiß, Sawyer et al. | 2022 | Historical Period Genomes from West Eurasia and the Mediterranean - Mapped reads | https://www.ebi.ac.uk/ena/browser/view/PRJEB53564 | EBI European Nucleotide Archive, PRJEB53564 |

The following previously published datasets were used:

| Author(s) | Year | Dataset title | Dataset URL | Database and Identifier |
|---|---|---|---|---|
| Moots et al. | 2022 | Mobility in the Iron Age Central Mediterranean | https://www.ebi.ac.uk/ena/browser/view/PRJEB49419 | EBI European Nucleotide Archive, PRJEB49419 |
| Mallick et al. | 2021 | Allen Ancient DNA Resource (AADR) v44.3 | https://doi.org/10.7910/DVN/FFIDCW | Harvard Dataverse, 10.7910/DVN/FFIDCW |

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

## Appendix 1

For each region, we show the results from pairwise individual-based qpAdm analysis in the form of heatmaps, annotated by dendrograms from the hierarchical clustering analysis. The heatmap color scheme is chosen to signify non-rejected hypotheses (above 0.05) in shades of red and orange, and rejected hypotheses in shades of gray and yellow.

Columns of the heatmaps are annotated by time period: Copper Age (CA), Bronze Age (BA), Iron Age (IA), Imperial Rome & Late Antiquity (IRLA), Middle Ages & Early Modern (MAEM).

Rows of the heatmaps are annotated by cluster assignment based on the hierarchical clustering (see Methods). Clusters identified as outliers through our pipeline downstream are also annotated on the rows. Please note that outlier detection was only performed for individuals from historical periods (IA, IRLA, MAEM).

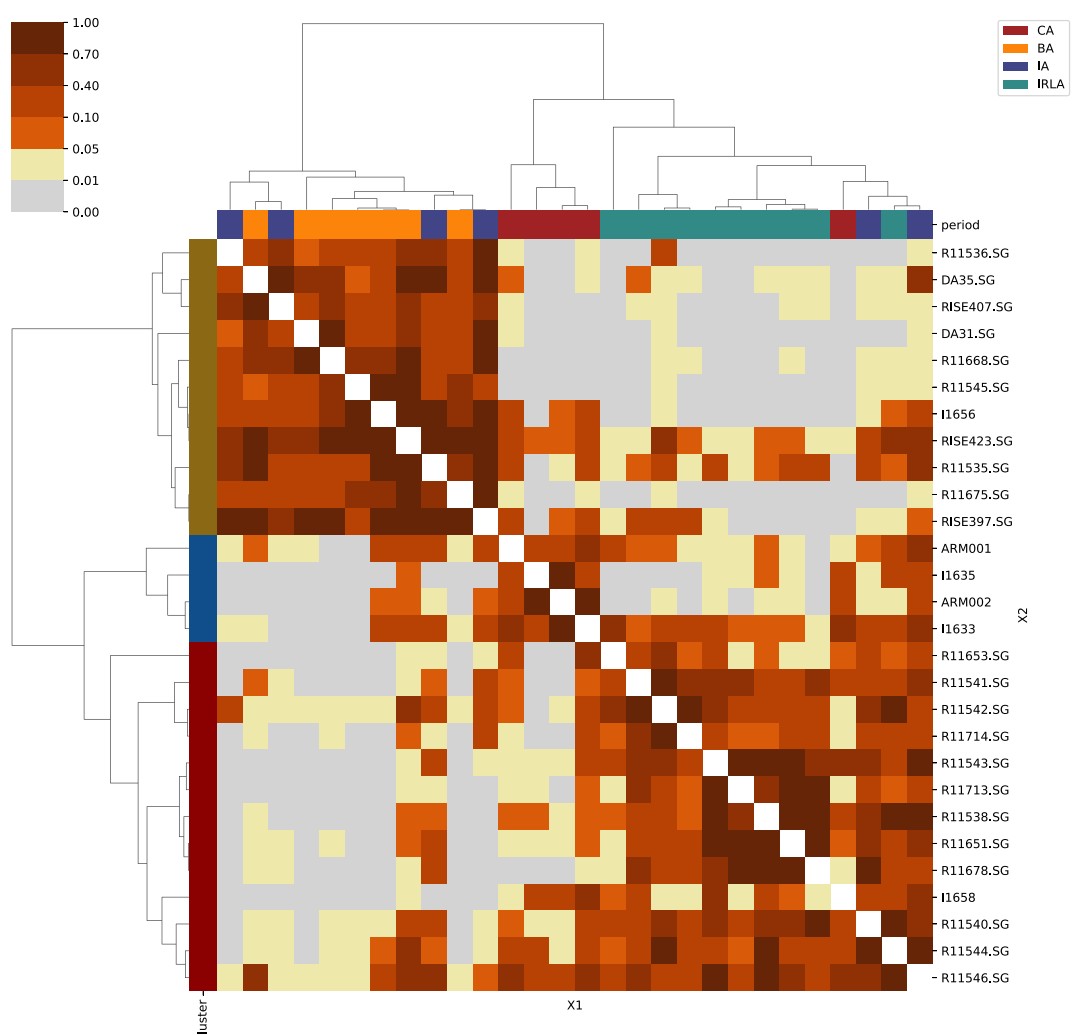

**Appendix 1—figure 1.** Armenia.

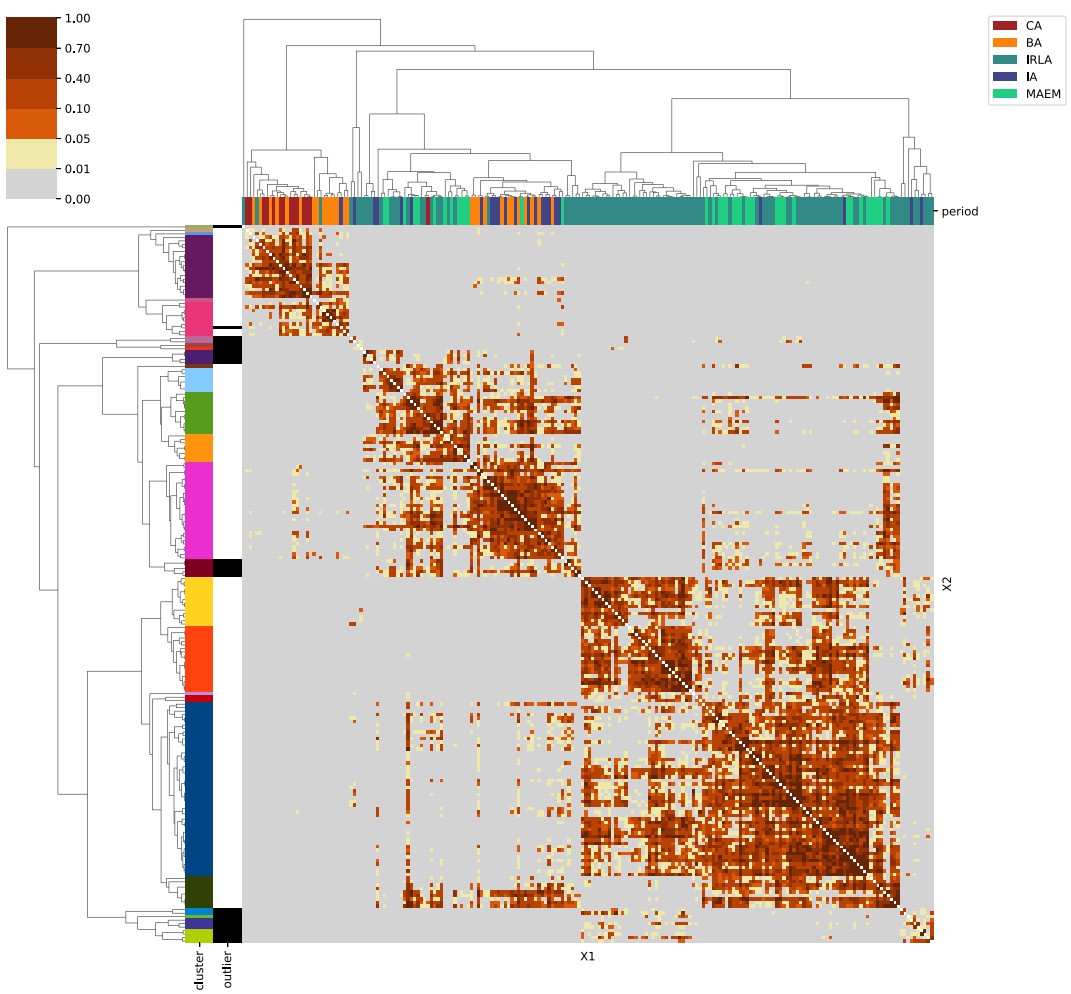

**Appendix 1—figure 2.** Mainland Italy.

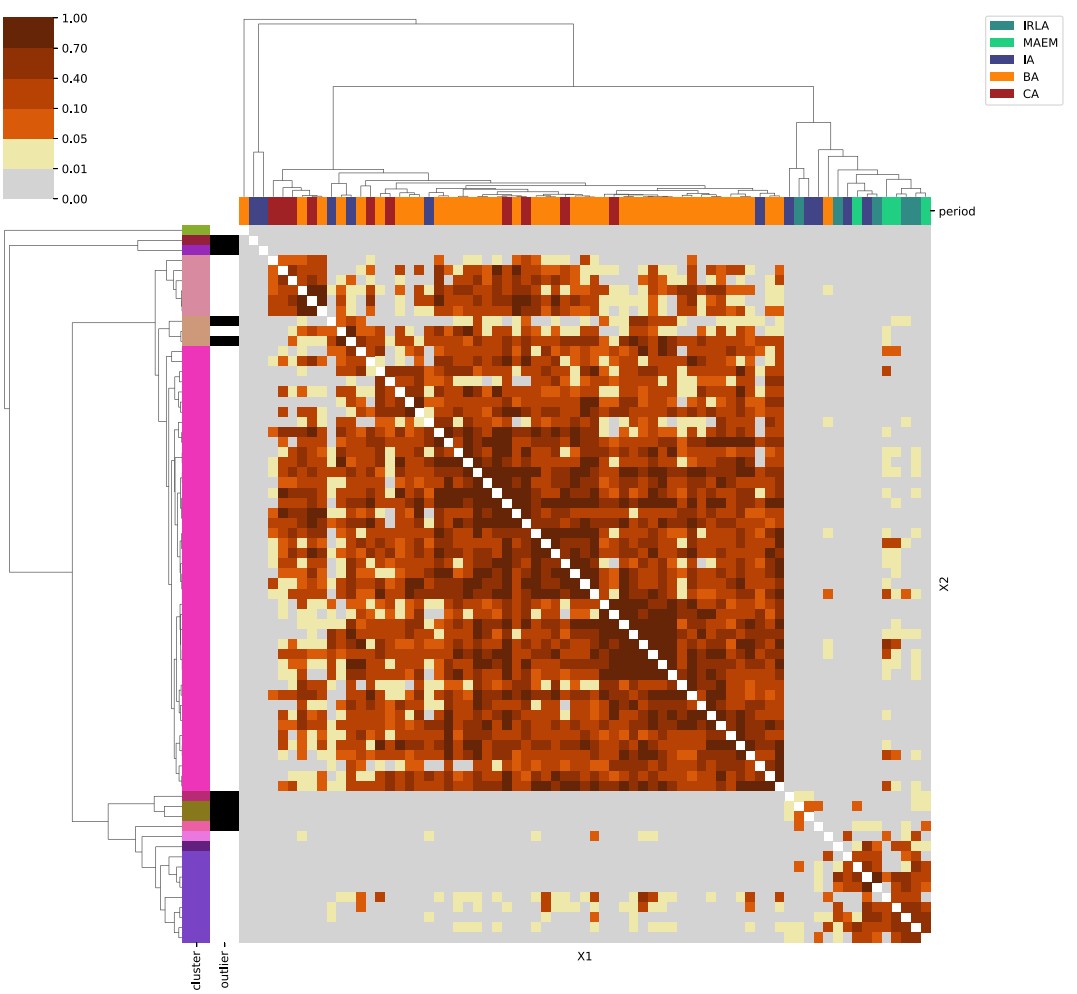

**Appendix 1—figure 3.** Sardinia.

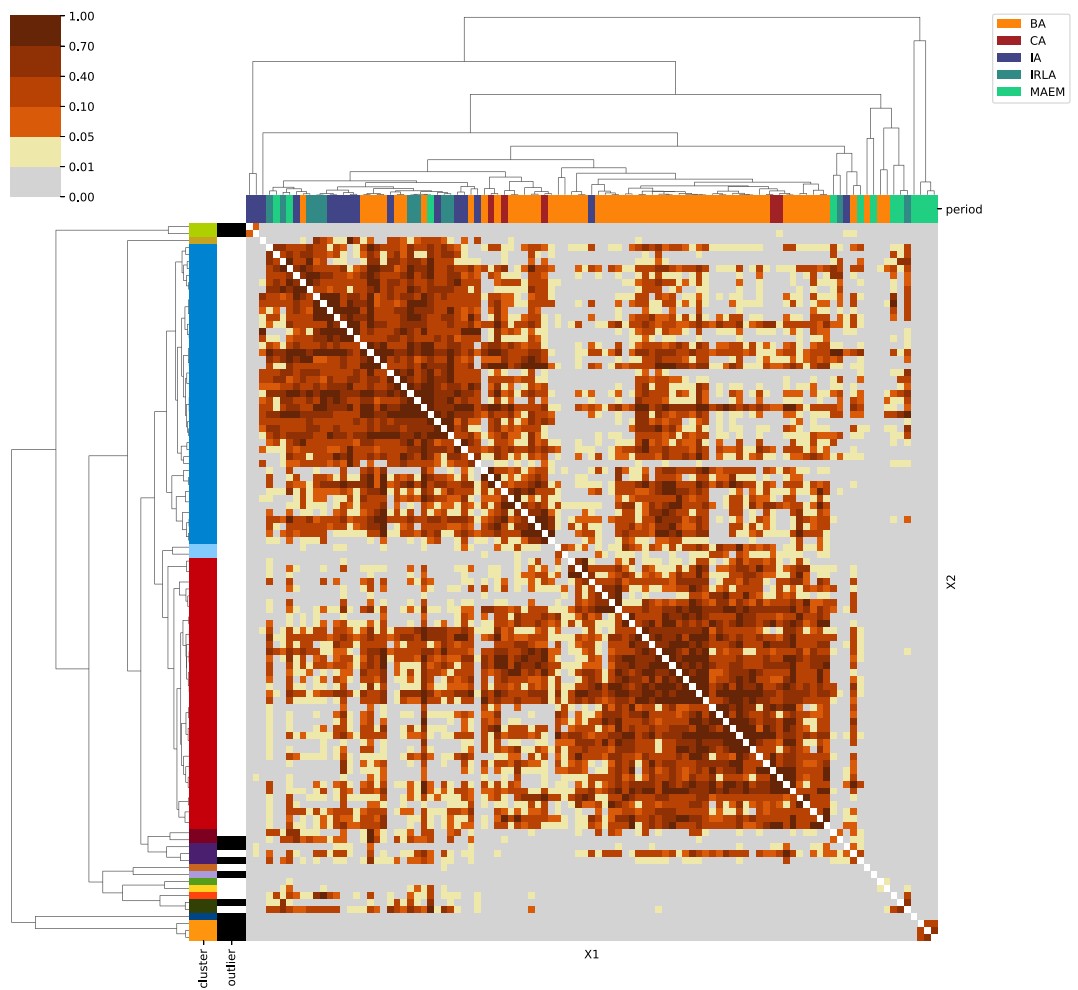

**Appendix 1—figure 4.** Levant & Egypt.

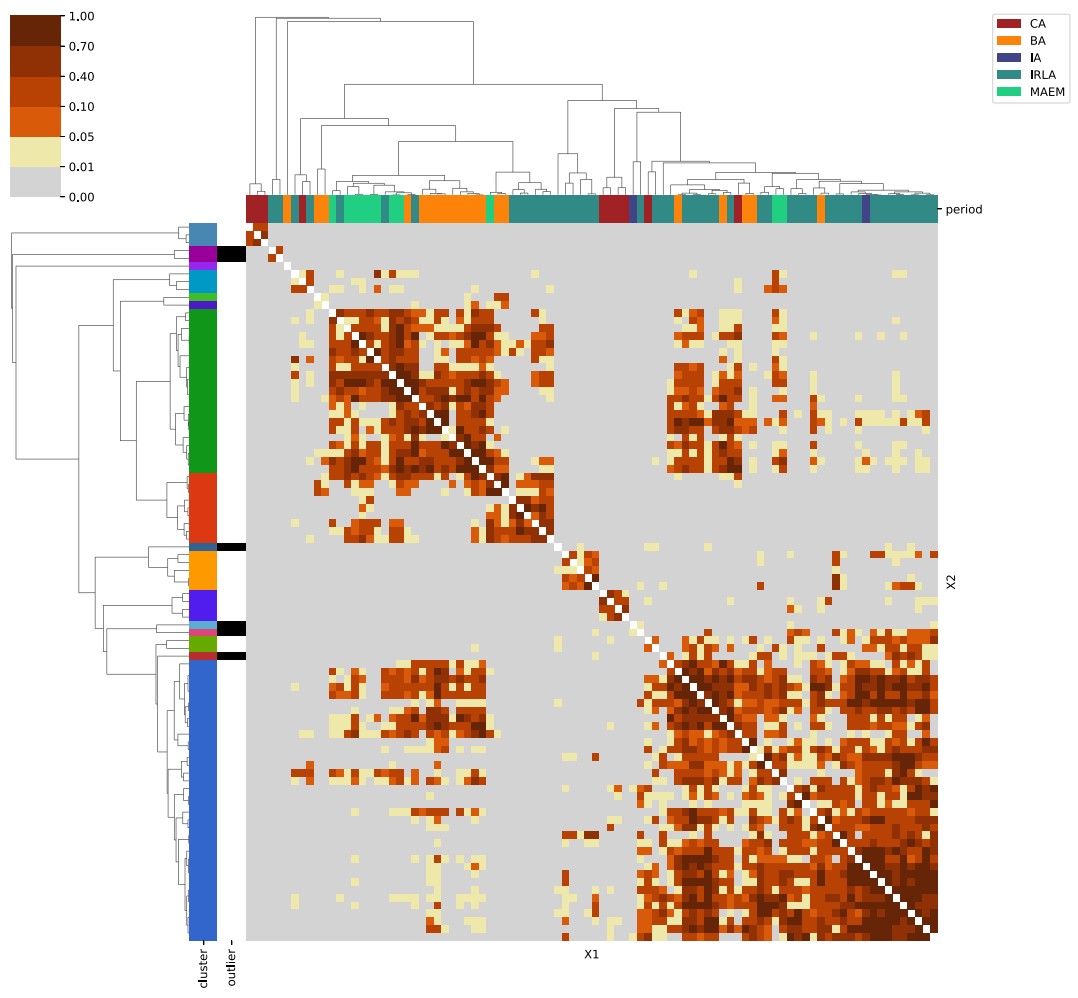

**Appendix 1—figure 5.** Southeastern Europe.

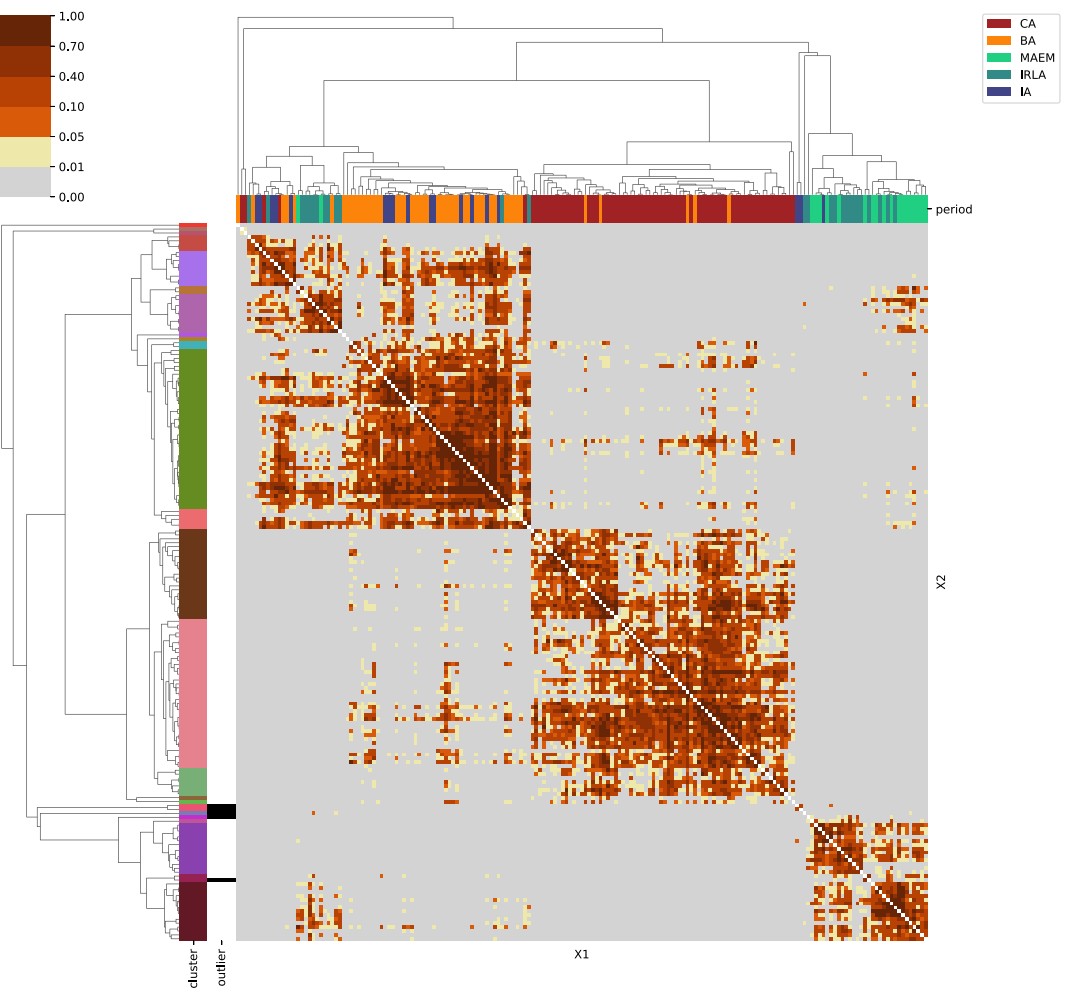

**Appendix 1—figure 6.** Southwestern Europe.

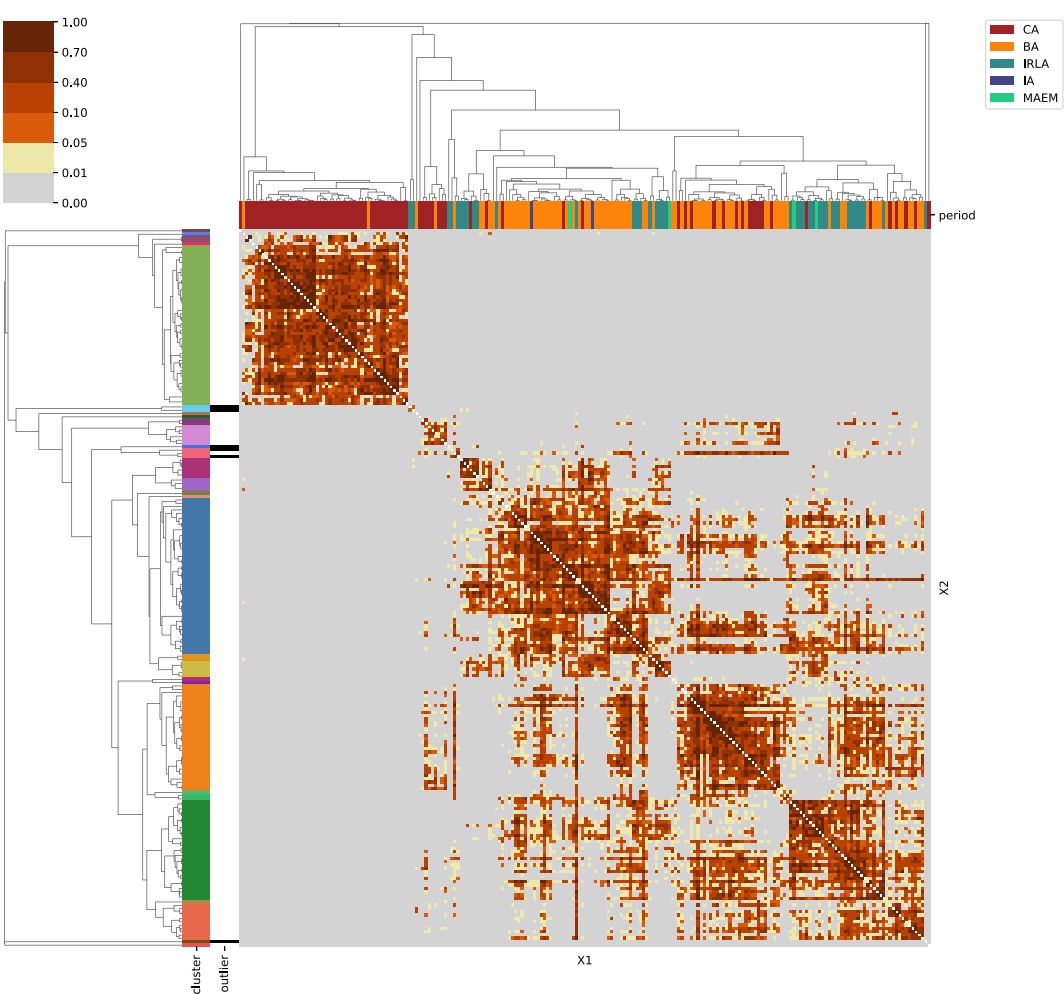

**Appendix 1—figure 7.** Western Europe.

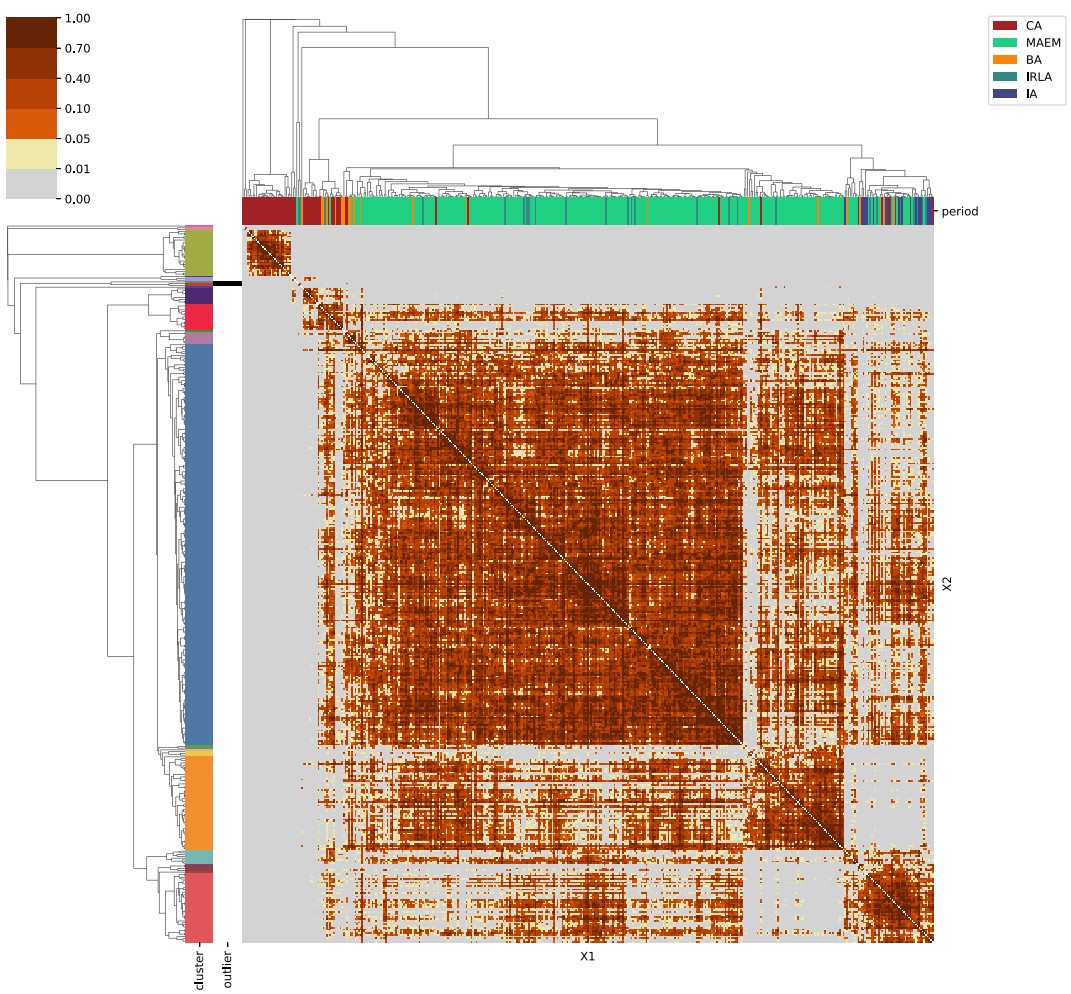

**Appendix 1—figure 8.** Northern Europe.

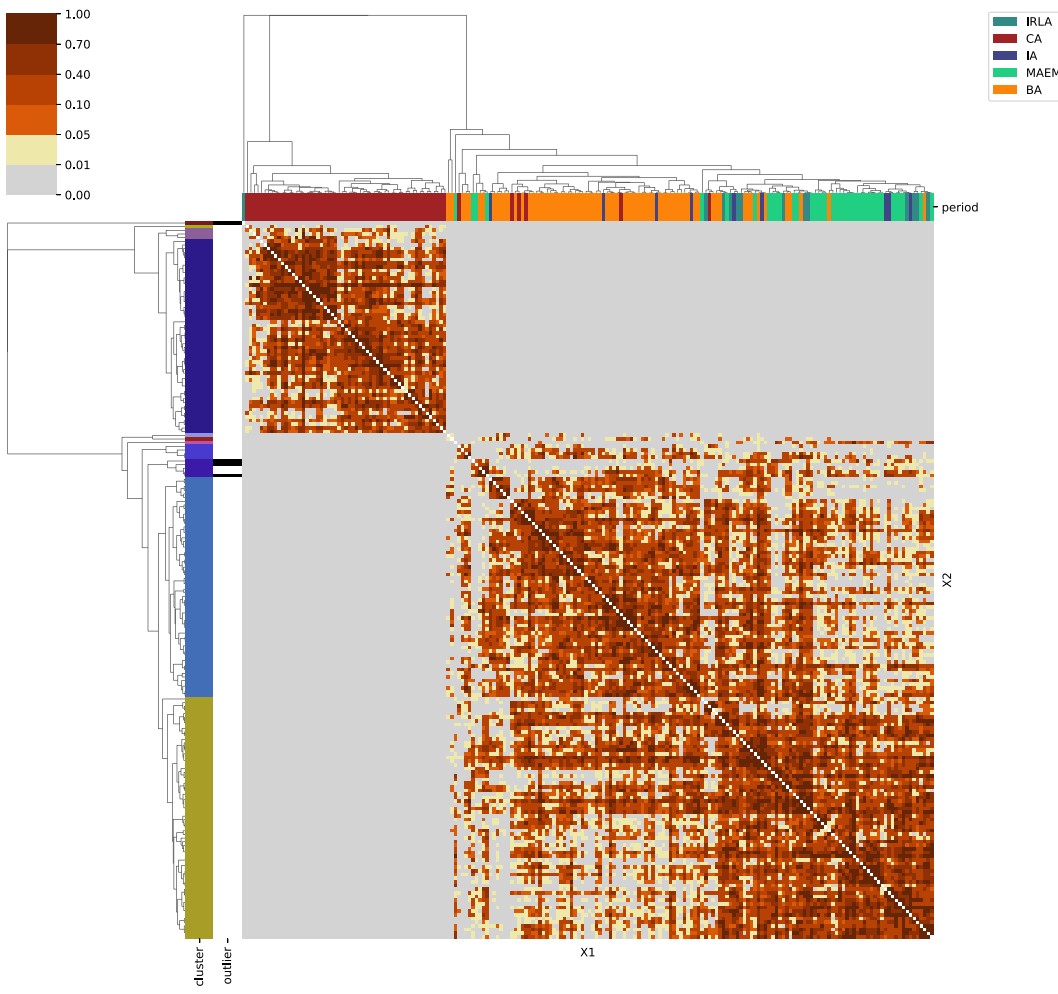

**Appendix 1—figure 9.** Great Britain & Ireland.

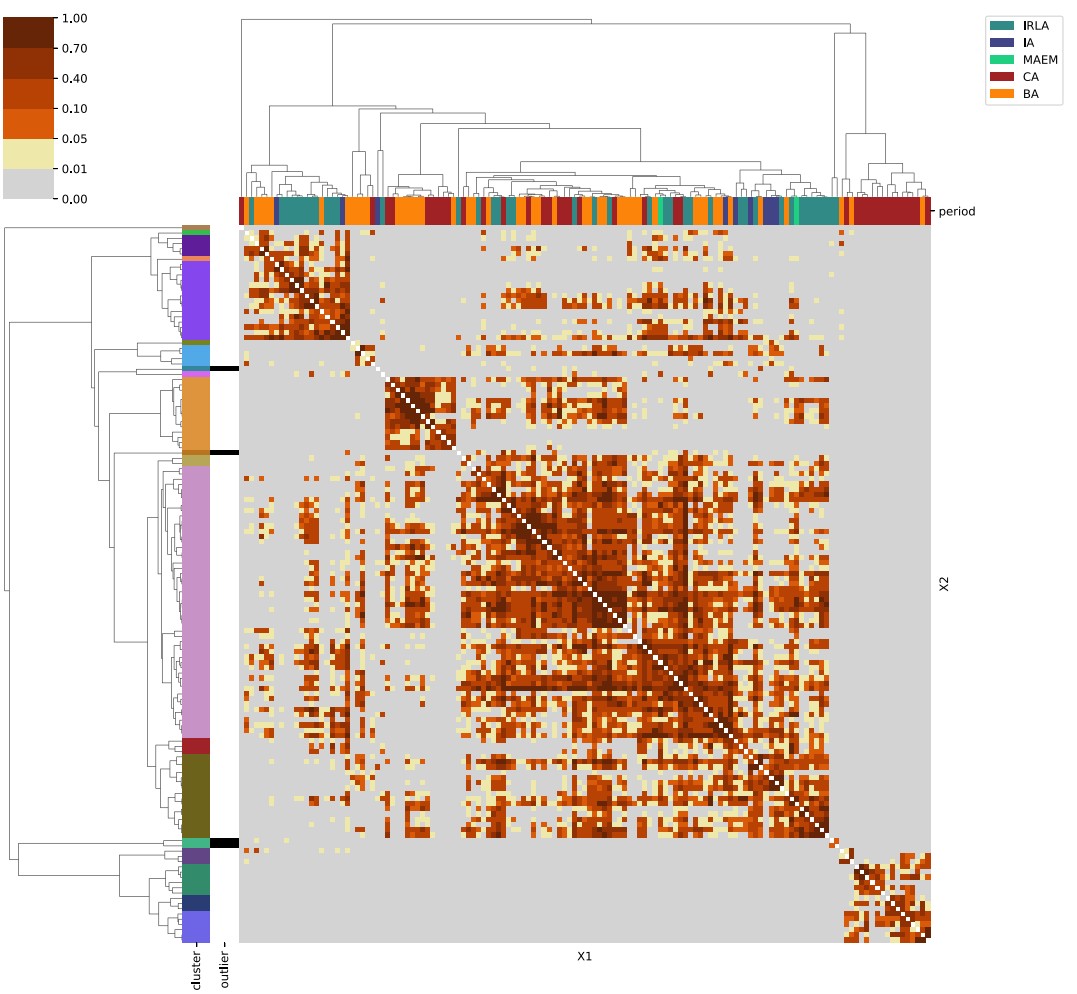

**Appendix 1—figure 10.** Eastern Central Europe.

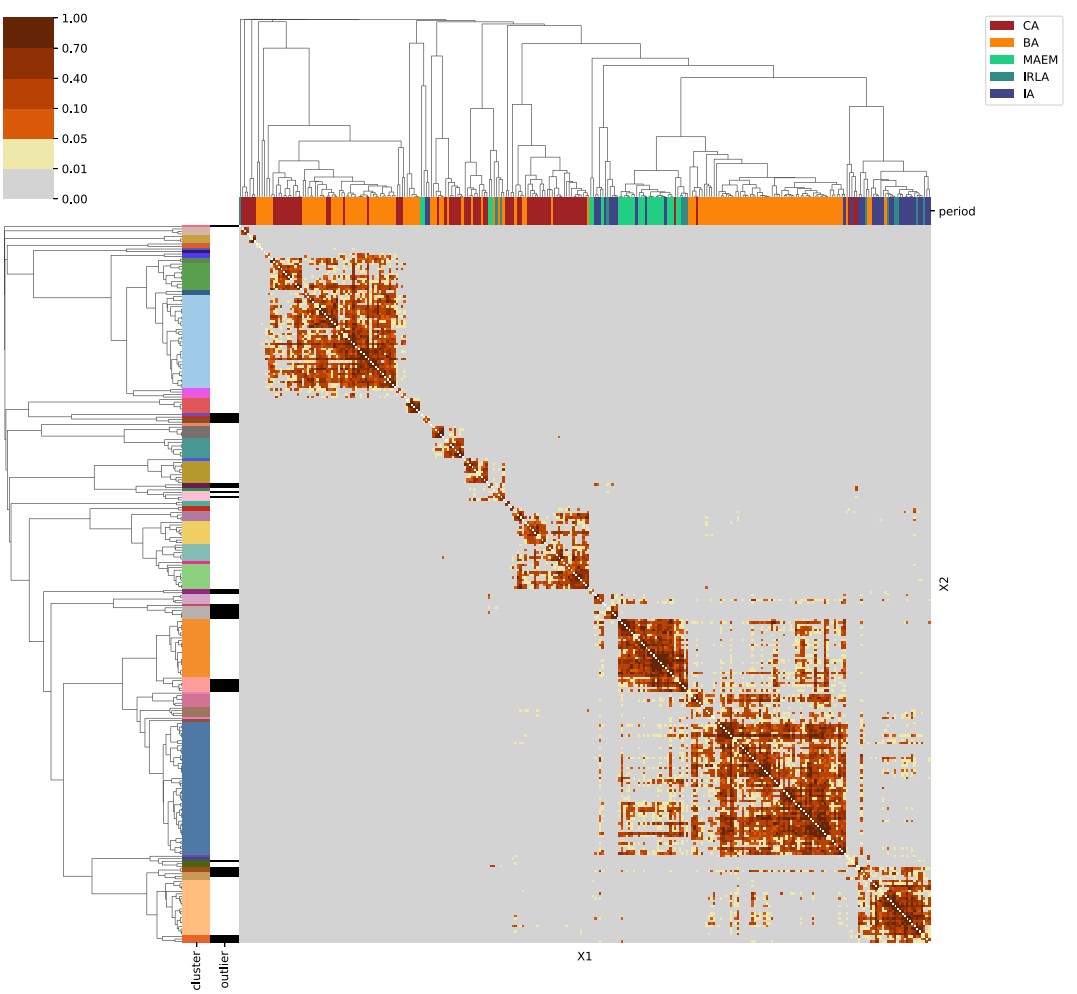

**Appendix 1—figure 11.** Eastern Europe & Steppe.

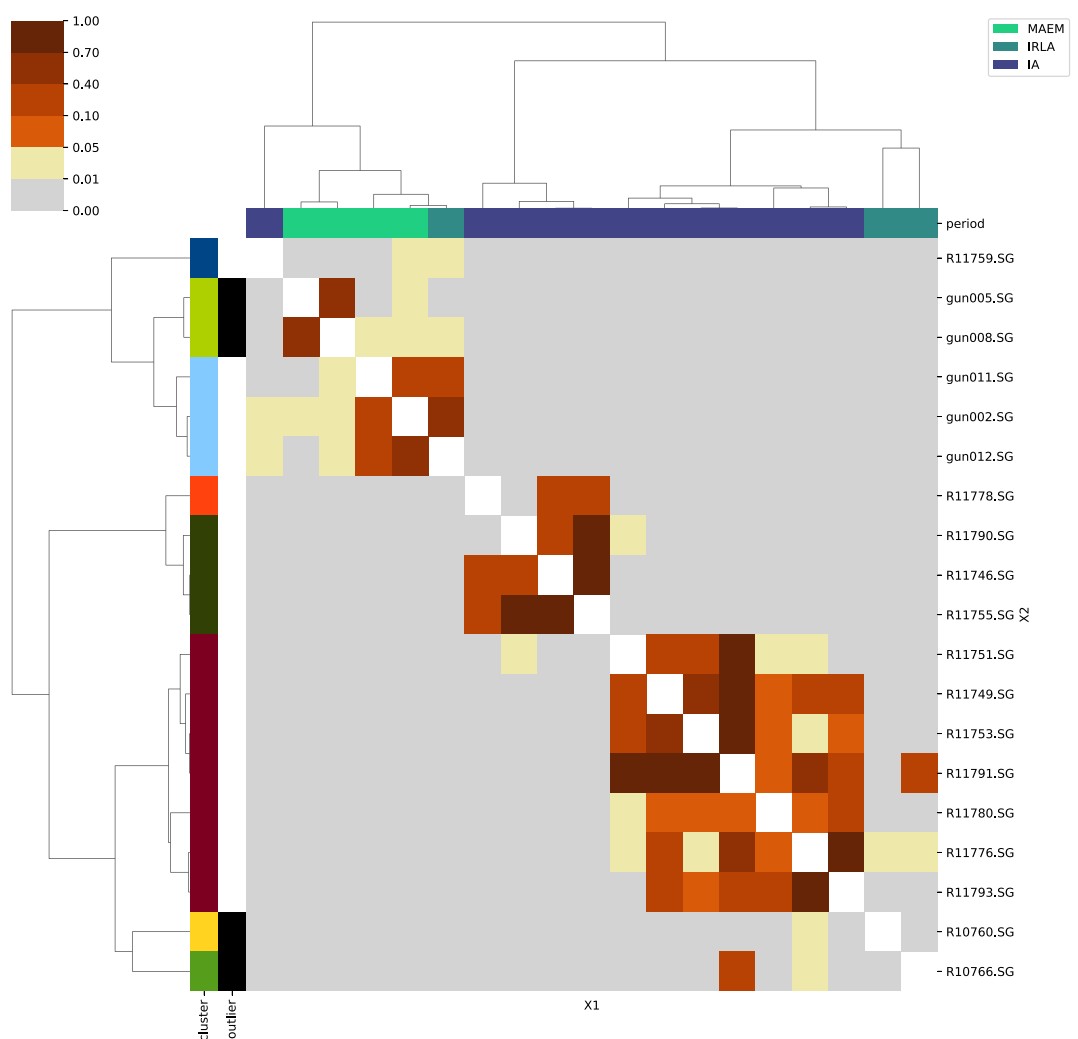

**Appendix 1—figure 12.** North Africa.

