## [Editor Report]

This important study provides an impressive dataset containing more than 200 novel ancient human genome sequences and a creative, robust, novel approach for studying human migration across time. The authors' conclusions are well supported by the data, and the methods used are convincing and solid. This paper will be of great interest to population geneticists and other scholars in the field of paleogenomics.

---

## [Decision Letter]

**Decision letter after peer review:**

Thank you for submitting your article "Stable population structure in Europe since the Iron Age, despite high mobility" for consideration by *eLife*. I am sorry it took a long time before we could get back to you with a full evaluation. Your article has been reviewed by 2 peer reviewers, and the evaluation has been overseen by a Reviewing Editor and Christian Landry as the Senior Editor. The following individual involved in review of your submission has agreed to reveal their identity: Benjamin Marco Peter (Reviewer #1).

Essential revisions:

1) Some technical concerns were raised regarding the statistical analyses. Some may require additional analyses and some need additional details on how the analyses were performed, including presenting the intermediate steps.

2) One reviewer mentioned that the origin of some of the data was difficult to trace.

*Reviewer #1 (Recommendations for the authors):*

This paper presents an interesting data set from historic Western Eurasia and North Africa. Overall, I commend the authors for presenting a comprehensive paper that focuses the data analysis of a large project on the major points, and that is easy to follow and well-written. Thus, I have no major comments on how the data was generated, or is presented. Paradoxically, historical periods are undersampled for ancient DNA, and so I think this data will be useful. The presentation is clever in that it focuses on a few interesting cases that highlight the breadth of the data.

The analysis is likewise innovative, with a focus on detecting "outliers" that are atypical for the genetic context where they were found. This is mainly achieved by using PCA and qpAdm, established tools, in a novel way. Here I do have some concerns about technical aspects, where I think some additional work could greatly strengthen the major claims made, and lay out if and how the analysis framework presented here could be applied in other work.

Clustering analysis

I have trouble following what exactly is going on here (particularly since the cited Fernandes et al. paper is also very ambiguous about what exactly is done, and doesn't provide a validation of this method). My understanding is the following: the goal is to test whether a pair of individuals (lets call them I1 and I2) are indistinguishable from each other, when we compare them to a set of reference populations. Formally, this is done by testing whether all statistics of the form F4(Ref_i, Ref_j; I1, I2) = 0, i.e. the difference between I1 and I2 is orthogonal to the space of reference populations, or that you test whether I1 and I2 project to the same point in the space of reference populations (which should be a subset of the PCA-space). Is this true? If so, I think it could be very helpful if you added a technical description of what precisely is done, and some validation on how well this framework works.

An independent concern is the transformation from p-values to distances. I am in particular worried about i) biases due to potentially different numbers of SNPs in different samples and ii) whether the resulting matrix is actually a sensible distance matrix (e.g. additive and satisfies the triangle inequality). To me, a summary that doesn't depend on data quality, like the F2-distance in the reference space (i.e. the sum of all F4-statistics, or an orthogonalized version thereof) would be easier to interpret. At the very least, it would be nice to show some intermediate results of this clustering step on at least a subset of the data, so that the reader can verify that the qpWave-statistics and their resulting p-values make sense.

The methodological concerns lead me to some questions about the data analysis. For example, in Figure 2, Supp 2, very commonly outliers lie right on top of a projected cluster. To my understanding, apart from using a different reference set, the approach using qpWave is equivalent to using a PCA-based clustering and so I would expect very high concordance between the approaches. One possibility could be that the differences are only visible on higher PCs, but since that data is not displayed, the reader is left wondering. I think it would be very helpful to present a more detailed analysis for some of these "surprising" clustering where the PCA disagrees with the clustering so that suspicions that e.g. low-coverage samples might be separated out more often could be laid to rest.

One way the presentation could be improved would be to be more consistent in what a suitable reference data set is. The PCAs (Figure 2, S1 and S2, and Fig6) argue that it makes most sense to present ancient data relative to present-day genetic variation, but the qpWave and qpAdm analysis compare the historic data to that of older populations. Granted, this is a common issue with ancient DNA papers, but the advantage of using a consistent reference data set is that the analyses become directly comparable, and the reader wouldn't have to wonder whether any discrepancies in the two ways of presenting the data are just due to the reference set.

PCA over time

It is a very interesting observation that the Fst-vs distance curve does not appear to change after the bronze age. However, I wonder if the comparison of the PCA to the projection could be solidified. In particular, it is not obvious to me how to compare Figure 6 B and C, since the data in C is projected onto that in Figure B, and so we are viewing the historic samples in the context of the present-day ones. Thus, to me, this suggests that ancient samples are most closely related to the folks that contribute to present-day people that roughly live in the same geographic location, at least for the middle east, north Africa and the Baltics, the three regions where the projections are well resolved.

Ideally, it would be nice to have independent PCAs (something F-stats based, or using probabilistic PCA or some other framework that allows for missingness). Alternatively, it could be helpful to quantify the similarity and projection error.

I have the following comments that pertain to the introduction.

Introduction:

I do agree that there is an ancient DNA sampling gap between the end of the Bronze age and the present day, and it is an interesting question to ask how bronze-age folks relate to present-day people. However, I don’t follow why the finding that present-day people are well-modelled as a three-way mixture of bronze-age groups would suggest demographic stability.

Within Western Eurasia, wouldn’t any demographic model with limited outside influences still maintain the principle of three-way mixtures? Do the authors mean to state that they believe that present day geographical genetic variation is mapping to the geographic genetic variation which existed ~6000 years ago? If so, then they should explicitly cite the studies and regions for which this has been established. In any case, this statement should be better motivated. Since this, to me, is an interesting finding of the present work (Figure 6) this might be better moved to the discussion.

*Reviewer #2 (Recommendations for the authors):*

Antonio, Weiss, Gao, Sawyer, et al. provide new ancient DNA (aDNA) data for 200 individuals from Europe and the Mediterranean from the historical period, including Iron Age, Late Antiquity, Middle Ages, and early modernity. These data are used to characterize population structure in Europe across time and identify first-generation immigrants (roughly speaking, those who present genetic ancestry that is significantly different from others in the same archaeological site). Authors provide an estimate of an average across regions of >8% of individuals being first-generation immigrants. This observation, coupled with the observed genetic heterogeneity across regions, suggests high mobility of individuals during the historical period in Europe. In spite of that, Principal Component Analysis (PCA) indicates that the overall population structure in Europe has been rather stable in the last 3,000 years, i.e., the levels of genetic differentiation across space have been relatively stable. To understand whether population structure stability is compatible with a large number (>8%) of long-distance immigrants, authors use spatially-explicit Wright-Fisher simulations. They conclude these phenomena are incompatible and provide a thoughtful and convincing explanation for that.

Overall I think this manuscript is very well written and provides an exciting take-home message. The dataset with 200+ novel ancient human genomes will be a great resource for population genetics and paleogenomic studies. Methods are robust and well-detailed. Although the methods used are well-known and standard in the field of paleogenomics, the way the authors use these methods is very creative, insightful, and refreshing. Results provide a comprehensive and novel assessment of historical population genetic structure in Europe, including characterizing genetic heterogeneity within populations and interactions/migration across regions. Conclusions are fully supported by the data.

A few of the strengths of this manuscript are its dataset containing a large number of ancient human genomes, the novel insights about human migration provided by the results, the creative approach to characterize migration and population structure across time using aDNA, and the excellent figures describing research results. I see no major issues with this paper.

1) I miss a table with the information regarding the published data, including sample size per region, per period, and source. On a related note, I miss the references to the papers that originally reported the public data being used in this manuscript. The manuscript cites AADR, but as per the AADR website, the individual papers that originally reported the data must be cited when AADR is used. It reads: Researchers who wish to use this compilation as the basis of their publications should cite this website and release version (e.g. "Allen Ancient DNA Resource [website]", version 50.0), while also citing the individual papers that report the data for each of the individuals they analyze. As an author of paper(s) that have originally reported data used in this manuscript, I believe we deserve to have our work acknowledged by users of the AADR compiled dataset and our papers cited. (forgive me if you have already done so and I missed it).

2) Figure 6C depicts the results of a PCA with ancient individuals projected onto present-day ones. Ancient (panel B) and present-day individuals (panel C) present similar spatial structures. I wonder if this is because ancient individuals were projected onto present-day ones. Have you tested whether a PCA with ancient individuals (perhaps a subset with a good breadth of coverage) also recovers the structure observed for present-day individuals? Forgive me if I missed it, but it would be nice to see a PCA plot for the three different historical periods showing the observed population structure is similar across time.

3) I don't see the number of variants observed in the simulations. I see that the simulated chromosome is 1e8 Mb long, but I don't see any mention of the number of observed variants. Perhaps I missed it – please let me know. I strongly suggest that you add this information to Figure 7 caption. I wonder if the different simulation scenarios could result in different numbers of variants being analyzed in each case and therefore different power to detect population structure using PCA (Figure 7).

4) Page 21, line 2: I'm surprised CpG sites are only 15% of the data. I had the impression that CpG sites accounted for ~2/3 of the 1240K dataset, but it's totally possible that you are correct. Please check if these numbers are accurate.

---

## [Author Response]

Essential revisions:1) Some technical concerns were raised regarding the statistical analyses. Some may require additional analyses and some need additional details on how the analyses were performed, including presenting the intermediate steps.2) One reviewer mentioned that the origin of some of the data was difficult to trace.

We thank the reviewers for their careful and positive reviews. We have done extensive work to address their comments, which has strengthened our results and the clarity of the paper. First, to address technical concerns about the statistical analyses, we revisited and updated our outlier detection pipeline and now provide additional detail on our technical procedures, both in the updated manuscript as well as in the response below. Additionally, intermediate results are reported in a new appendix document. Almost all figures were updated as a result of this, and we believe these updates have strengthened our results, especially regarding the cross-regional connections our pipeline identified, while our main findings still stand. Second, to address the reviewer’s comments regarding the origin of published data, we added a supplementary file that lists all published samples that contributed to this study as well as citations of the primary sources. Lastly, we thank the reviewers for comments on the clarity and accessibility of the manuscript, which we believe has substantially improved through these revisions.

Reviewer #1 (Recommendations for the authors):This paper presents an interesting data set from historic Western Eurasia and North Africa. Overall, I commend the authors for presenting a comprehensive paper that focuses the data analysis of a large project on the major points, and that is easy to follow and well-written. Thus, I have no major comments on how the data was generated, or is presented. Paradoxically, historical periods are undersampled for ancient DNA, and so I think this data will be useful. The presentation is clever in that it focuses on a few interesting cases that highlight the breadth of the data.The analysis is likewise innovative, with a focus on detecting "outliers" that are atypical for the genetic context where they were found. This is mainly achieved by using PCA and qpAdm, established tools, in a novel way. Here I do have some concerns about technical aspects, where I think some additional work could greatly strengthen the major claims made, and lay out if and how the analysis framework presented here could be applied in other work.Clustering analysisI have trouble following what exactly is going on here (particularly since the cited Fernandes et al. paper is also very ambiguous about what exactly is done, and doesn't provide a validation of this method). My understanding is the following: the goal is to test whether a pair of individuals (lets call them I1 and I2) are indistinguishable from each other, when we compare them to a set of reference populations. Formally, this is done by testing whether all statistics of the form F4(Ref_i, Ref_j; I1, I2) = 0, i.e. the difference between I1 and I2 is orthogonal to the space of reference populations, or that you test whether I1 and I2 project to the same point in the space of reference populations (which should be a subset of the PCA-space). Is this true? If so, I think it could be very helpful if you added a technical description of what precisely is done, and some validation on how well this framework works.

We agree that the previous description of our workflow was lacking, and have substantially improved the description of the entire pipeline (Methods, section “Modeling ancestry and identifying outliers using qpAdm”), making it clearer and more descriptive. To further improve clarity, we have also unified our use of methodology and replaced all mentions of “qpWave” with “qpAdm”. In the reworked Methods section mentioned above, we added a discussion on how these tests are equivalent in certain settings, and describe which test we are exactly doing for our pairwise individual comparisons, as well as for all other qpAdm tests downstream of cluster discovery. In addition, we now include an additional appendix document (Appendix 1) which, for each region, shows the results from our individual-based qpAdm analysis and clustering in the form of heatmaps, in addition to showing the clusters projected into PC space.

An independent concern is the transformation from p-values to distances. I am in particular worried about i) biases due to potentially different numbers of SNPs in different samples and ii) whether the resulting matrix is actually a sensible distance matrix (e.g. additive and satisfies the triangle inequality). To me, a summary that doesn't depend on data quality, like the F2-distance in the reference space (i.e. the sum of all F4-statistics, or an orthogonalized version thereof) would be easier to interpret. At the very least, it would be nice to show some intermediate results of this clustering step on at least a subset of the data, so that the reader can verify that the qpWave-statistics and their resulting p-values make sense.

We agree that calling the matrix generated from p-values a “distance matrix” is a misnomer, as it does not satisfy the triangle inequality, for example. We still believe that our clustering generates sensible results, as UPGMA simply allows us to project a positive, symmetric matrix to a tree, which we can then use, given some cut-off, to define clusters. To make this distinction clear, we now refer to the resulting matrix as a “dissimilarity matrix” instead. As mentioned above, we now also include a supplementary figure for each region visualizing the clustering results.

Regarding the concerns about p-values conflating both signal and power, we employ a stringent minimum SNP coverage filter for these analyses to avoid extremely-low coverage samples being separated out (min. SNPs covered: 100,000). In addition, we now show that cluster size and downstream outlier status do not depend on SNP coverage (Figure 2 —figure supplement 3).

The methodological concerns lead me to some questions about the data analysis. For example, in Figure 2, Supp 2, very commonly outliers lie right on top of a projected cluster. To my understanding, apart from using a different reference set, the approach using qpWave is equivalent to using a PCA-based clustering and so I would expect very high concordance between the approaches. One possibility could be that the differences are only visible on higher PCs, but since that data is not displayed, the reader is left wondering. I think it would be very helpful to present a more detailed analysis for some of these "surprising" clustering where the PCA disagrees with the clustering so that suspicions that e.g. low-coverage samples might be separated out more often could be laid to rest.

To reduce the risk of artifactual clusters resulting from our pipeline, we devised a set of QC metrics (described in detail below) on the individuals and clusters we identified as outliers. Driven by these metrics, we implemented some changes to our outlier detection pipeline that we now describe in substantially more detail in the Methods (see comment above). Since the pipeline involves running many thousands of qpAdm analyses, it is difficult to manually check every step for all samples – instead, we focused our QC efforts on the outliers identified at the end of the pipeline. To assess outlier quality we used the following metrics, in addition to manual inspection:

First, for an individual identified as an outlier at the end of the pipeline, we check its fraction of non-rejected hypotheses across all comparisons within a region. The rationale here is that by definition, an outlier shouldn’t cluster with many other samples within its region, so a majority of hypotheses should be rejected (corresponding to gray and yellow regions in the heatmaps, Appendix 1). Through our improvements to the pipeline, the fraction of non-rejected hypotheses was reduced from an average of 5.3% (median 1.1%) to an average of 3.8% (median 0.6%), while going from 107 to 111 outliers across all regions.

Second, we wanted to make sure that outlier status was not affected by the inclusion of pre-historic individuals in our clustering step within regions. To represent majority ancestries that might have been present in a region in the past, we included Bronze and Copper Age individuals in the clustering analysis. We found that including these individuals in the pairwise analysis and clustering improved the clusters overall. However, to ensure that their inclusion did not bias the downstream identification of outliers, we also recalculated the clustering without these individuals. We inspected whether an individual identified as an outlier would be part of a majority cluster in the absence of Bronze and Copper Age individuals, which was not the case (see also the updated Methods section for more details on how we handle time periods within regions).

In response to the “surprising” outliers based on the PCA visualizations in Figure 2, Supplement 2: with our updated outlier pipeline, some of these have disappeared, for example in Western and Northern Europe. However, in some regions the phenomenon remains. We are confident this isn’t a coverage effect, as we’ve compared the coverage between outliers and non-outliers across all clusters (see previous comment, Figure 2 —figure supplement 3), as well as specifically for “surprising” outliers compared to contemporary non-outliers – none of which showed any differences in the coverage distributions of “surprising” outliers (Author response images 1 and 2). In addition, we believe that the quality metrics we outline above were helpful in minimizing artifactual associations of samples with clusters, which could influence their downstream outlier status. As such, we think it is likely that the qpAdm analysis does detect a real difference between these sets of samples, even though they project close to each other in PCA space. This could be the result of an actual biological difference hidden from PCA by the differences in reference space (see also the reply to the following comment). Still, we cannot fully rule out the possibility of latent technical biases that we were not able to account for, so we do not claim the outlier pipeline is fully devoid of false positives. Nevertheless, we believe our pipeline is helpful in uncovering true, recent, long-range dispersers in a high-throughput and automated manner, which is necessary to glean this type of insight from hundreds of samples across a dozen different regions.

**Author response image 1. sa2fig1:** SNP coverage comparison between outliers and non-outliers in region-period pairings with “surprising” outliers (t-test p-value: 0. 242).

**Author response image 2. sa2fig2:** PCA projection (left) and SNP coverage comparison (right) for “surprising” outliers and surrounding non-outliers in Italy_IRLA.

One way the presentation could be improved would be to be more consistent in what a suitable reference data set is. The PCAs (Figure 2, S1 and S2, and Fig6) argue that it makes most sense to present ancient data relative to present-day genetic variation, but the qpWave and qpAdm analysis compare the historic data to that of older populations. Granted, this is a common issue with ancient DNA papers, but the advantage of using a consistent reference data set is that the analyses become directly comparable, and the reader wouldn't have to wonder whether any discrepancies in the two ways of presenting the data are just due to the reference set.

While it is true that some of the discrepancies are difficult to interpret, we believe that both views of the data are valuable and provide complementary insights. We considered three aspects in our decision to use both reference spaces: (1) conventions in the field (including making the results accessible to others), (2) interpretability, and (3) technical rigor.

Projecting historical genomes into the present-day PCA space allows for a convenient visualization that is common in the field of ancient DNA and exhibits an established connection to geographic space that is easy to interpret. This is true especially for more recent ancient and historical genomes, as spatial population structure approaches that of present day. However, there are two challenges: (1) a two-dimensional representation of a fairly high-dimensional ancestry space necessarily incurs some amount of information loss and (2) we know that some axes of genetic variation are not well-represented by the present-day PCA space. This is evident, for example, by projecting our qpAdm reference populations into the present-day PCA, where some ancestries which we know to be quite differentiated project closely together (Author response image 3). Despite this limitation, we continue to use the PCA representation as it is well resolved for visualization and maximizes geographical correspondence across Eurasia.

On the other hand, the qpAdm reference space (used in clustering and outlier detection) has higher resolution to distinguish ancestries by more comprehensively capturing the fairly high-dimensional space of different ancestries. This includes many ancestries that are not well resolved in the present-day PCA space, yet are relevant to our sample set, for example distinguishing Iranian Neolithic ancestry against ancestries from further into central and east Asia, as well as distinguishing between North African and Middle Eastern ancestries (Author response image 3).

To investigate the differences between these two reference spaces, we chose pairwise outgroup-f3 statistics (to Mbuti) as a pairwise similarity metric representing the reference space of f-statistics and qpAdm in a way that’s minimally affected by population-specific drift. We related this similarity measure to the euclidean distance on the first two PCs between the same set of populations (Author response image 4). This analysis shows that while there is almost a linear correspondence between these pairwise measures for some populations, others comparisons fall off the diagonal in a manner consistent with PCA projection (Author response image 3), where samples are close together in PCA but not very similar according to outgroup-f3. Taken together, these analyses highlight the non-equivalence of the two reference spaces.

In addition, we chose to base our analysis pipeline on the f-statistics framework to (1) afford us a more principled framework to disentangle ancestries among samples and clusters within and across regions (using 1-component vs. 2-component models of admixture), while (2) keeping a consistent, representative reference set for all analyses that were part of the primary pipeline. Meanwhile, we still use the present-day PCA space for interpretable visualization.

**Author response image 3. sa2fig3:** Projection of qpAdm reference population individuals into present-day PCA.

**Author response image 4. sa2fig4:** Comparison of pairwise PCA projection distance to outgroup-f3 similarity across all qpAdm reference population individuals. PCA projection distance was calculated as the euclidean distance on the first two principal components. Outgroup-f3 statistics were calculated relative to Mbuti, which is itself also a qpAdm reference population. Both panels show the same data, but each point is colored by either of the two reference populations involved in the pairwise comparison.

PCA over timeIt is a very interesting observation that the Fst-vs distance curve does not appear to change after the bronze age. However, I wonder if the comparison of the PCA to the projection could be solidified. In particular, it is not obvious to me how to compare Figure 6 B and C, since the data in C is projected onto that in Figure B, and so we are viewing the historic samples in the context of the present-day ones. Thus, to me, this suggests that ancient samples are most closely related to the folks that contribute to present-day people that roughly live in the same geographic location, at least for the middle east, north Africa and the Baltics, the three regions where the projections are well resolved.Ideally, it would be nice to have independent PCAs (something F-stats based, or using probabilistic PCA or some other framework that allows for missingness). Alternatively, it could be helpful to quantify the similarity and projection error.

The fact that historical period individuals are “most closely related to the folks that contribute to present-day people that roughly live in the same geographic location” is exactly the point we were hoping to make with Figures 6 B and C. We do realize, however, that the fact that one set of samples is projected into the PC space established by the other may suggest that this is an obvious result. To make it more clear that it is not, we added an additional panel to Figure 6, which shows pre-historical samples projected into the present-day PC space. This figure shows that pre-historical individuals project all across the PCA space and often outside of present-day diversity, with degraded correlation of geographic location and projection location (see also Author response image 5). This illustrates the contrast we were hoping to communicate, where projection locations of historical individuals start to “settle” close to present-day individuals from similar geographic locations, especially in contrast with pre-historic individuals.

**Author response image 5. sa2fig5:** Comparing geographic distance to PCA distance between pairs of historical and pre-historical individuals matched by geographic space. For each historical period individual we selected the closest pre-historical individual by geographic distance in an effort to match the distributions of pairwise geographic distance across the two time periods (left). For these distributions of individuals matched by geographic distance, we then queried the euclidean distance between their projection locations in the first two principal components (right).

I have the following comments that pertain to the introduction.Introduction:I do agree that there is an ancient DNA sampling gap between the end of the Bronze age and the present day, and it is an interesting question to ask how bronze-age folks relate to present-day people. However, I don’t follow why the finding that present-day people are well-modelled as a three-way mixture of bronze-age groups would suggest demographic stability.Within Western Eurasia, wouldn’t any demographic model with limited outside influences still maintain the principle of three-way mixtures? Do the authors mean to state that they believe that present day geographical genetic variation is mapping to the geographic genetic variation which existed ~6000 years ago? If so, then they should explicitly cite the studies and regions for which this has been established. In any case, this statement should be better motivated. Since this, to me, is an interesting finding of the present work (Figure 6) this might be better moved to the discussion.

We agree with the reviewer that the logic seemed somewhat circular here, as we use a finding from our study as motivation in the introduction. We have rephrased this section to be more clearly stated as a question we intended to study with this work, rather than a motivating previous result.

Reviewer #2 (Recommendations for the authors):Antonio, Weiss, Gao, Sawyer, et al. provide new ancient DNA (aDNA) data for 200 individuals from Europe and the Mediterranean from the historical period, including Iron Age, Late Antiquity, Middle Ages, and early modernity. These data are used to characterize population structure in Europe across time and identify first-generation immigrants (roughly speaking, those who present genetic ancestry that is significantly different from others in the same archaeological site). Authors provide an estimate of an average across regions of >8% of individuals being first-generation immigrants. This observation, coupled with the observed genetic heterogeneity across regions, suggests high mobility of individuals during the historical period in Europe. In spite of that, Principal Component Analysis (PCA) indicates that the overall population structure in Europe has been rather stable in the last 3,000 years, i.e., the levels of genetic differentiation across space have been relatively stable. To understand whether population structure stability is compatible with a large number (>8%) of long-distance immigrants, authors use spatially-explicit Wright-Fisher simulations. They conclude these phenomena are incompatible and provide a thoughtful and convincing explanation for that.Overall I think this manuscript is very well written and provides an exciting take-home message. The dataset with 200+ novel ancient human genomes will be a great resource for population genetics and paleogenomic studies. Methods are robust and well-detailed. Although the methods used are well-known and standard in the field of paleogenomics, the way the authors use these methods is very creative, insightful, and refreshing. Results provide a comprehensive and novel assessment of historical population genetic structure in Europe, including characterizing genetic heterogeneity within populations and interactions/migration across regions. Conclusions are fully supported by the data.A few of the strengths of this manuscript are its dataset containing a large number of ancient human genomes, the novel insights about human migration provided by the results, the creative approach to characterize migration and population structure across time using aDNA, and the excellent figures describing research results. I see no major issues with this paper.1) I miss a table with the information regarding the published data, including sample size per region, per period, and source. On a related note, I miss the references to the papers that originally reported the public data being used in this manuscript. The manuscript cites AADR, but as per the AADR website, the individual papers that originally reported the data must be cited when AADR is used. It reads: Researchers who wish to use this compilation as the basis of their publications should cite this website and release version (e.g. "Allen Ancient DNA Resource [website]", version 50.0), while also citing the individual papers that report the data for each of the individuals they analyze. As an author of paper(s) that have originally reported data used in this manuscript, I believe we deserve to have our work acknowledged by users of the AADR compiled dataset and our papers cited. (forgive me if you have already done so and I missed it).

We apologize for the oversight! We now include an additional supplementary file

(Supplementary File 4) that includes all published samples that contributed to this study, and have added additional relevant citations.

2) Figure 6C depicts the results of a PCA with ancient individuals projected onto present-day ones. Ancient (panel B) and present-day individuals (panel C) present similar spatial structures. I wonder if this is because ancient individuals were projected onto present-day ones. Have you tested whether a PCA with ancient individuals (perhaps a subset with a good breadth of coverage) also recovers the structure observed for present-day individuals? Forgive me if I missed it, but it would be nice to see a PCA plot for the three different historical periods showing the observed population structure is similar across time.

Please see our response to a similar comment by Reviewer 1, which we hope addresses this comment.

3) I don't see the number of variants observed in the simulations. I see that the simulated chromosome is 1e8 Mb long, but I don't see any mention of the number of observed variants. Perhaps I missed it – please let me know. I strongly suggest that you add this information to Figure 7 caption. I wonder if the different simulation scenarios could result in different numbers of variants being analyzed in each case and therefore different power to detect population structure using PCA (Figure 7).

We agree that this is an important point, and have added the number of variants contributing to the PCA of our simulation results to the “Simulations” Methods section. They were roughly equivalent to the numbers of variants used for the data analysis, so we think our analyses are similarly powered.

4) Page 21, line 2: I'm surprised CpG sites are only 15% of the data. I had the impression that CpG sites accounted for ~2/3 of the 1240K dataset, but it's totally possible that you are correct. Please check if these numbers are accurate.

The sites we removed from our dataset were based on a list of SNPs that are transitions at CpG sites. We clarified the wording in the methods.